# SCALING DENSITIES FOR IMPROVED DENSITY RATIO ESTIMATION

## ABSTRACT

Estimating the discrepancy between two densities ($p$ and $q$) is central to machine learning. Most frequently used methods for the quantification of this discrepancy capture it as a function of the ratio of the densities $p/q$. In practice, closed-form expressions for these densities or their ratio are rarely available. As such, estimating density ratios accurately using only samples from $p$ and $q$ is of high significance and has led to a flurry of recent work in this direction. Among these, binary classification based density ratio estimators have shown great promise and have been extremely successful in specialized domains. However, estimating the density ratio using a binary classifier, when the samples from the densities are *well separated*, remains challenging. In this work, we first show that the state-of-the-art solutions for such *well separated* cases have limited applicability, may suffer from theoretical inconsistencies or lack formal guarantees and therefore perform poorly in the general case. We then present an alternative framework for density ratio estimation that is motivated by the scaled-Bregman divergence. Our proposal is to scale the densities $p$ and $q$ by another density $m$ and estimate $\log p/q$ as $\log p/m - \log q/m$. We show that if the scaling measures are constructed such that they overlap with $p$ and $q$, then a single multi-class logistic regression can be trained to accurately recover $p/m$ and $q/m$ on samples from $p, q$ and $m$. We formally justify our method with the scaled-Bregman theorem and show that it does not suffer from the issues that plague the existing solutions. We provide a large battery of empirical evaluations of our method with both synthetic and real datasets on the tasks of density ratio estimation, mutual information estimation, and representation learning. Finally, we demonstrate that our method can be applied to improve non-classification based model-free density ratio estimators as well.

## 1 INTRODUCTION

Quantification of discrepancy between two distributions underpins the foundation of a large number of machine learning techniques. Of especial prominence are distribution discrepancy measures known as $f$-divergences (Csiszár, 1964), which are defined as expectations of the functions of the ratio of the two densities. As such, density ratio estimation is often a central task in deep generative modeling, mutual information and divergence estimation, representation learning, etc. (Sugiyama et al., 2012; Gutmann & Hyvärinen, 2010; Goodfellow et al., 2014; Nowozin et al., 2016; Srivastava et al., 2017; Belghazi et al., 2018; Oord et al., 2018; Srivastava et al., 2020). However, estimating density ratio by modeling each of the densities is challenging for most problems of interest as modeling high dimensional densities is a harder problem than estimating their ratios Sugiyama et al. (2012). Therefore, in practice, estimators are employed that only require samples from the pair of densities in order to estimate their ratio without explicitly modeling the individual densities. One of the most commonly used density ratio estimators (DRE) is a binary classifier based DRE (BC-DRE) which can be formally shown to estimate the ground truth density ratio when trained to correctly discriminate be-

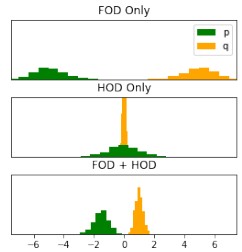

Figure 1: First Order Discrepancy vs Higher Order Discrepancy.

tween the samples from the two densities (e.g. Gutmann & Hyvärinen, 2010; Gutmann & Hirayama, 2011; Sugiyama et al., 2012; Menon & Ong, 2016).

BC-DREs have been tremendously successful in problems involving minimization of the discrepancy between the data and the model distributions even in high-dimensional settings (Nowozin et al., 2016; Radford et al., 2015). However, they do not fare as well when applied to the task of estimating the discrepancy between two distributions that are far apart or easily separable otherwise. This issue has been characterized recently as the *density-chasm problem* by Rhodes et al. (2020). The discrepancy between two distributions can manifest itself in many different ways. Here, we roughly group these ways into two classes, **First Order Discrepancy (FOD)** and **Higher Order Discrepancy (HOD)**. Two distributions can easily be separated when the first order statistics, such as their means, are far apart from each other. We refer to this type of discrepancy between $p$ and $q$ as FOD. Similarly, if the discrepancy between $p$ and $q$ (with no FOD) originates due to the difference in higher order statistics we refer to this as HOD. For example, consider Figure 1 where in all the three cases, the KL$[p\|q] = 50$, but in (a) it originates from FOD since $\mathcal{N}(\mu_1 = -5, \sigma_1 = 1), \mathcal{N}(\mu_1 = 5, \sigma_1 = 1)$, in (b) it originates from HOD since $\mathcal{N}(\mu_1 = 0, \sigma_1 = 1), \mathcal{N}(\mu_2 = 0, \sigma_1 = 0.097)$, and in (c) it originates from FOD and HOD since $\mathcal{N}(\mu_1 = -1.5, \sigma_1 = 0.5), \mathcal{N}(\mu_2 = 1, \sigma_1 = 0.25)$.

In a recent work, Choi et al. (2021) attempt to address the density-chasm problem. They considered distributions $p$ and $q$ that are clearly separated and proposed that in such cases density ratios can be better estimated if the mixture of the two densities is mapped onto a unit-sphere using a flow-based map (Rezende & Mohamed, 2015). Using a bijective map ensures that the density ratio between the original distributions is maintained post mapping. While this method (F-DRE) may resolve the density-chasm problem due to FOD, we found that it struggles when the densities are separated due to HOD.

On the other hand, Rhodes et al. (2020) proposed to estimate the density ratio between $p$ and $q$ as $\frac{p}{p_1} \times \frac{p_1}{p_2} \times \cdots \times \frac{p_K}{q}$. This telescoping is achieved by training $K$ BC-DREs where the distributions $p_1, \ldots, p_k$ are designed to be *closely-packed* such that all of the $K$ ratios can be easily estimated using BC-DREs without suffering from the density-chasm problem. They call this estimator, telescoping density ratio estimator or TRE. Empirically, we found that while TRE performs relatively better than BC-DRE and F-DRE in cases with density-chasm problem due to HOD, it cannot deal with the density-chasm problem originating due to FOD. Intuitively, this happens because TRE trains $K$ DREs, each of which is trained on $K$ different pairs of distributions. This leads to the distribution shift issue at test time, especially when $p$ and $q$ have high FOD. Further, care needs to be taken when designing intermediate distributions $p_1, \ldots, p_K$ to ensure that each of the ratios are well defined. In fact, the construction schemes introduced in the original work may render TRE theoretically ill-defined if both $p$ and $q$ do not have full support (Appendix A).

We will show that neither of these state-of-the-art methods are applicable when the density ratio estimation problem involves both FOD and HOD. Unfortunately, most problems in practice tend to have both types of discrepancy. This leaves a clear technical gap in the task of density ratio estimation. Hence, in this work, we present SCALED DENSITY RATIO ESTIMATOR (sDRE), a novel method for estimating density ratio that:

1. Theoretically well-defined and does not suffer from distribution shift.
2. Resolves both FOD and HOD types of density-chasm issues.
3. Does not require training additional FLOW-based models or BC-DREs and therefore is computationally more efficient.

## 2 NOTATION AND BACKGROUND

Bregman distance between two $d$-dimensional vectors $a$ and $b$ is defined as $B_\phi(a, b) = \phi(a) - \phi(b) - \nabla\phi(b)(a - b)$. Here $\phi : \mathbb{R}^d \mapsto \mathbb{R}$ is a convex function. Following the notation in Stummer & Vajda (2012) the scaled-Bregman Divergence (sBD) between probability measures $P$ and $Q$ scaled by an arbitrary measure $M$ can be defined as,

$$B_f(P, Q|M) = \int_{\mathcal{X}} m\left[ f\left(\frac{p}{m}\right) - f\left(\frac{q}{m}\right) - f'\left(\frac{q}{m}\right)\left(\frac{p}{m} - \frac{q}{m}\right) \right] d\lambda. \tag{1}$$

Here, $f : \mathcal{X} \to \mathbb{R}$ is a convex function, $f'$ is its right derivative. Further, all the measures $P, Q$ and $M$ are defined on the measurable space $(\mathcal{X}, \mathcal{A})$ and are absolutely continuous with respect to a $\sigma$-finite measure $\lambda$, therefore admitting the following densities,

$$p = \frac{dP}{d\lambda}, \quad q = \frac{dQ}{d\lambda}, \quad m = \frac{dM}{d\lambda}. \tag{2}$$

It is easy to show that for $f(t) = t \log(t)$, sBD reduces to KL divergence and is independent of the scaling measure $M$ and $\lambda$,

$$B_{t \log t}(P, Q | M) = \int_{\mathcal{X}} m \left[ \frac{p}{m} \log \left( \frac{p}{m} \right) - \frac{p}{m} \log \left( \frac{q}{m} \right) \right] d\lambda = \text{KL}(P \| Q). \tag{3}$$

We define

$$r_{p/m}(x) = \frac{p(x)}{m(x)}, \quad r_{q/m}(x) = \frac{q(x)}{m(x)}, \quad r_{p/q}(x) = \frac{p(x)}{q(x)} \tag{4}$$

as the density ratios between $p$ & $m$, $q$ & $m$ and $p$ & $q$ respectively.

Given samples $\{x_p^i\}_{i=1}^N$ and $\{x_q^j\}_{j=1}^M$ from distributions $p$ and $q$ respectively and assuming $p << q$, the goal of DRE $\hat{r}_{p/q}$ is to estimate $r_{p/q}$.

## 2.1 BINARY CLASSIFICATION TO DRE USING BD

Proposition 3 (Menon & Ong, 2016) formally establishes the link between binary classification and density ratio estimation using sBD. We re-state it here.

**Proposition 1 (Proposition 3 (Menon & Ong, 2016) rephrased)** *Given a class-probability estimator $\hat{\eta} : \mathcal{X} \mapsto [0, 1]$ such that the density ratio estimator, $\hat{r}_{p/q}(x) = \frac{\hat{\eta}}{1 - \hat{\eta}}$. Then for any convex differentiable $f : [0, 1] \mapsto \mathbb{R}$,*

$$\mathbb{E}_{x \sim p_{marginal}}[D_f(\eta(X) \| \hat{\eta}(X)] = \frac{1}{2} \mathbb{E}_{x \sim q}[D_{f^\dagger}(r_{p/q}(X) \| \hat{r}_{p/q}(X))] \tag{5}$$

*where $D_f$ is Bregman Divergence (BD), $f^\dagger : [0, \infty) \mapsto \mathbb{R} \mid f^\dagger(x) = (1 + x) f(\frac{x}{1+x})$, $D_{f^\dagger}$ is sBD and $p_{marginal}$ is the marginal distribution of $X$.*

## 2.2 DENSITY CHASM

Kato & Teshima (2021) note that most empirical estimators of density ratio, especially those implemented using a deep neural network tend to over-fit the loss function in some way or the other. This phenomenon was recently characterized by Rhodes et al. (2020) for the BC-DRE as the density chasm problem. They found that the BC-DRE tends to substantially underestimate the ground-truth KL-divergence (as a function of the estimated ratio) between $p$ and $q$ when these distributions are significantly separated from each other. We conjecture that this underestimation occurs because in the case of finite training data, when the two sets of samples are easily separable, the estimated ratios do not need to be as large as they theoretically had to be to achieve perfect classification performance. In such a case, many different decision boundaries achieve equally good performance.

## 3 RELATED WORK

Our work is most closely related to and improves upon the shortcomings of the recently proposed TRE method. Therefore, in order to better understand these shortcomings, we begin by analyzing TRE under Proposition 3 of Menon & Ong (2016) as described in equation 5 above.

TRE uses a two step, divide-and-conquer strategy to resolve the density chasm problem. In the first step, they construct $K$ *waymark* distributions $p_1, \ldots, p_K$ by *gradually* transporting samples from $p$ towards samples from $q$. Then, they train $K$ BC-DRE models, one for each consecutive pairs of distributions. This allows for estimating the ratio $r_{p/q}$ as the product of the $K$ BC-DREs,

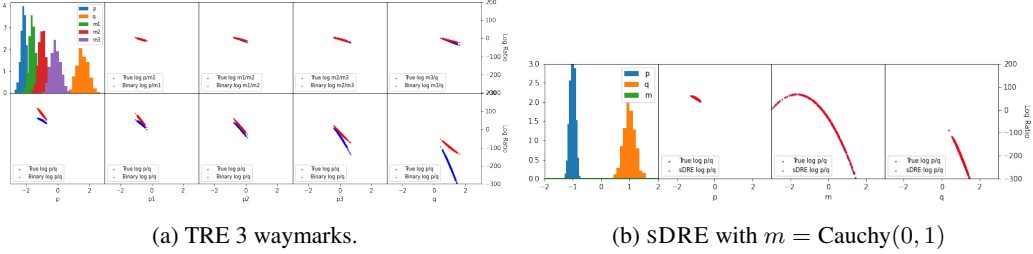

(a) TRE 3 waymarks.   (b) sDRE with $m = \text{Cauchy}(0, 1)$

Figure 2: TRE vs sDRE on $p = \mathcal{N}(-1, 0.1)$ and $q = \mathcal{N}(1, 0.2)$

$\frac{p}{q} = \frac{p}{p_1} \times \cdots \times \frac{p_K}{q}$. They introduced two schemes for creating waymark distributions that make sure that consecutive pairs of distributions are packed *close-enough* to ensure that none of the $K$ BC-DREs suffers from the density-chasm issue. Hence, TRE attempts to resolve the density-chasm issue by replacing the ratio between $p$ and $q$ with a *bridge* of $K$ intermediate density ratios that by design (of the waymark distribution) do not suffer from density-chasm problem.

Application of Proposition 3 to TRE shows that each of the $K$ individual binary classifiers, upon training lead to $K$ DREs that minimize the sBD to their corresponding true density ratios on the samples from their corresponding denominator densities. This however, does not guarantee that a product of such individual DREs minimizes the sBD to $r_{p/q}$ on samples from $q$ (or all $K$ intermediate distributions for $K > 1$). This issue is acknowledged by the authors who note that TRE's performance suffers significantly from the distribution shift issue especially when the distributions $p$ and $q$ have high FOD. An example of this can be seen in Figure 2a where we train TRE to estimate the density ratio between $p = \mathcal{N}(-1, 0.1)$ and $q = \mathcal{N}(+1, 0.2)$ using 3 intermediate distributions $p_1, p_2, p_3$ constructed using the *linear-mixing* construction described in Section 4.2. In the first row we evaluate the ratios $p/p_1, p_1/p_2, p_2/p_3$ and $p_3/q$ on samples from their respective denominator densities and in the second row we evaluate $\log \frac{p}{q}$ on samples from all the $K + 2$ distributions. As per equation 5, individual DREs are in fact able to estimate their corresponding density ratios accurately on the samples from their corresponding denominator densities. But, their combination does not lead to an accurate estimation of $p/q$ for any of the $K + 2$ distributions except $p_2$, making TRE not useful for estimating discrepancy measures such as KL divergence as the model is not accurate on samples from $p$.

Another interesting related work comes from Choi et al. (2021) (F-DRE), who propose to train BC-DRE in a learned feature space where the two distributions $p$ and $q$ are mapped onto a unit sphere. To this end, they employ a FLOW-based model on the input and train it to map the mixture of the samples from the two distributions to a standard Gaussian. It is easy to show that any bijective map will preserve the original density ratio $r_{p/q}$ in the feature space as the Jacobian correction term simply cancels out. However, such a method cannot *bring the distributions closer* along both FOD and HOD since doing that will change the original density ratio. Due to this, in practice, we found that F-DRE tends to work only in fairly limited settings.

## 4 DENSITY RATIO ESTIMATION BY SCALING DENSITIES

We tackle the density chasm problem using the notion of scaling densities from the scaled-Bregman divergence (equation 1, sBD). sBD introduces a scaling measure $m$ in the estimation of the discrepancy between two distributions $p$ and $q$. This allows for changing the sampling distribution in the estimation of, for instance, KL divergence as described in equation 3 from $p$ to $m$ by re-writing the original integrand $\log \frac{p}{q}$ as

$$\log \frac{p}{q} = \log \frac{p}{m} - \log \frac{q}{m}. \tag{6}$$

We leverage this insight and propose the SCALED DENSITY RATIO ESTIMATOR (sDRE) method for density ratio estimation that is theoretically well-defined, computationally more efficient, does not suffer from distribution-shift issue and thereby, mitigates the density-chasm problem originating from both FOD and HOD discrepancies. To this end, in sDRE, we first scale the densities $p$ and $q$

by another probability measure $m$ and estimate $\log \frac{p}{q}$ as $\log \frac{p}{m} - \log \frac{q}{m}$. If the scaling measure $m$ is constructed such that $p << m$, $q << m$ and its samples overlap with the samples from both $p$ and $q$, then as shown in the next section, a single multi-class logistic regression trained to classify samples from $p, q$ and $m$ into 3 classes, can accurately recover $\frac{p}{m}$ and $\frac{q}{m}$ on samples from any of the 3 distributions. Later, we show that the sDRE approach can also be applied to model-free DREs (Section 5.3).

## 4.1 MULTI-CLASS CLASSIFICATION TO MULTI-DISTRIBUTION DRE

Nock et al. (2016) build upon Proposition 3 of Menon & Ong (2016) (equation 5) using their scaled-Bregmann theorem and state the following lemma.

**Lemma 1** *Given a class-probability estimator* $\hat{\eta} : \mathcal{X} \mapsto [0,1]^{C-1}$*, let the density ratio estimator* $\hat{r}(x) = \frac{\hat{\eta}(x)}{\hat{\eta}_C(x)}$*. Then for any convex differentiable* $f : [0,1]^{C-1} \mapsto \mathbb{R}$*,*

$$\mathbb{E}_{x \sim p_{marginal}}[D_f(\eta(X) \| \hat{\eta}(X))] = (1 - \pi_C)\mathbb{E}_{x \sim p_C}[D_{f^\dagger}(r(X) \| \hat{r}(X))]. \tag{7}$$

Here, $f^\dagger = g(x)f\left(\frac{x}{g(x)}\right)$ and $g(x) = \frac{\pi_C}{1 - \pi_C} + \tilde{\pi}^T x$. $\pi_C$ is the prior probability of class $C$, $\eta_C(x)$ is the class probability of $x$ under the class C and $p_C$ is the distribution of X indexed by the class $C$. $\hat{r}$ is a $C - 1$ dimensional vector of density ratios between the $C - 1$ densities $(p_1, \ldots p_{C-1})$ and the reference density $p_C$. Similarly, $\hat{\eta}(x)$ is a $C - 1$-dimensional vector of class probabilities. Please note that the choice of the reference class $C$ is arbitrary. In simple terms, Lemma 1 shows that a multi-class logistic regression (among other multi-class classifiers) leads to a multi-class density ratio estimator that minimizes the sBD to the corresponding true density ratio on samples from the reference distribution.

We demonstrate the implication of **Lemma 1** on sDRE in Figure 2b. Here, we train sDRE on the same exact setup as TRE in Figure 2a where $p = \mathcal{N}(-1, 0.1)$ and $q = \mathcal{N}(+1, 0.2)$. To this end, we propose to use $\text{Cauchy}(0, 1)$ as the scaling measure. This is because the samples from the heavy tailed distributions easily overlap with the highly concentrated samples of $p$ and $q$ as shown in Figure 2b. Then we train a single 3-way logistic regression to classify samples from $p, q$ and $m$. As clearly shown, sDRE is able to perfectly estimate $\log p/q$ on samples from all the three distributions without suffering from any distribution shift issues. This empirically confirms that, unlike TRE, the sDRE estimator converges to the correct density ratio on the entire domain of $m$ as long as $m$ is such that both $p$ and $q$ are absolutely continuous to $m$.

sDRE can also use more than one scaling measures $\{m_k\}_{k=1}^K$. Not only does Lemma 1 hold true for any number $(K)$ of distributions, but also Ma & Collins (2018) show that, when the number of samples is fixed, increasing $K$ leads to a more consistent estimator. Please note, increasing the number of scaling measure does not change the estimation of $r_{p/q}$. In a multi-class logistic regression model trained on samples from $p, q$ and $m_k$, the ratio between $p$ and $q$ is estimated as the function of their logits $h_1, h_2, h_k$ as follows. From Lemma 1, $r_{p/m_k} = \frac{\exp(h_1)}{\exp(h_k)}$ i.e. $\log r_{p/m_k} = \log \frac{\exp(h_1)}{\exp(h_k)} = h_1 - h_k$. Similarly, $\log r_{q/m_k} = \log \frac{\exp(h_2)}{\exp(h_k)} = h_2 - h_k$. This gives us the final ratio as, $\log r_{p/q} = \log r_{p/m_k} - \log r_{q/m_k} = h_1 - h_2$, that can be calculated similarly with respect to any of the $K$ scaling measure.

To demonstrate that the improved performance of sDRE over TRE in the example from Figure 2 does not only stem from the novel construction scheme for $m$, we also train sDRE using TRE's waymark distributions as scaling measures. This is illustrated in Figure 5c in Appendix B. Unlike TRE that does not estimate $\log \frac{p}{q}$ accurately on samples from $p, q, p_1$ and $p_3$, sDRE is able to estimate $\log \frac{p}{q}$ on all 5 distributions significantly better even with sub-optimal scaling measures.

## 4.2 CONSTRUCTING M

In this work, we use three ways of constructing the scaling measure $M$. Here we describe them briefly. Please see Appendix A for details. **Overlapping Distribution:** We introduce a novel scaling measure construction scheme under which, $M$ is defined as any distribution whose samples overlap with both $p$ and $q$ and $p << m$, $q << m$. For example, it can be heavy-tailed

distributions, such as Cauchy and Student-t, or simply normal and uniform distributions or their mixtures. **Linear Mixing:** In this construction scheme, distribution $M$ is defined as the empirical distribution of the samples constructed by linearly combining samples $X_p = \{x_p^i\}_{i=1}^N$ and $X_q = \{x_q^i\}_{i=1}^N$ from distributions $p$ and $q$ respectively. That is, $m$ is the empirical distribution over the set $X_m = \{x_m^i | x_m^i = (1 - \alpha)x_p^i + \alpha x_q^i, x_p \in X_p, x_q^i \in X_q\}_{i=1}^N$. This construction is related to the linear combination waymark of Rhodes et al. (2020). We clarify the difference in Appendix A. **Dimension-wise Mixing:** In this construction scheme, that is borrowed from TRE as it is, $M$ is defined as the empirical distribution of the samples generated by combining different subsets of dimensions from samples from $p$ and $q$.

### 4.3 sDRE vs TRE

sDRE differs from TRE in two major ways. First, TRE instructs to create $K$, *closely-packed* waymark distributions $p_1, \ldots, p_K$ between $p$ and $q$ such that none of the $K+1$ BC-DREs that it proposes to train, suffers from the density chasm problem. This approach is not only computationally more demanding as it requires training $K+1$ models (with some parameter sharing), but also leads to the aforementioned distribution shift issue. In contrast, sDRE instructs to introduce scaling densities $\{m_k\}_{k=1}^K$ such that their samples overlap with the samples from both $p$ and $q$. It then estimates $\log \frac{p}{q}$ as $\log \frac{p}{m_k} - \log \frac{q}{m_k}$ by training a *single* multi-class logistic regression model across all $K+2$ distributions. This approach is not only computationally more efficient, but is also theoretically guaranteed to not suffer from distribution shift as shown in the previous section.

Second, for $K = 1$, TRE and sDRE are similar as they both can be seen as using telescoping of ratios to estimate $r_{p/q}$. However, they are not equivalent. TRE proposes the following telescoping: $\log p/q = \log p/m + \log m/q$ and sDRE proposes to telescope in the following manner: $\log p/q = \log p/m - \log q/m$. As such, TRE and sDRE use two different sets ($\frac{dM}{dQ}$ vs $\frac{dQ}{dM}$ in the second term) of Radon-Nikodym Derivatives (RND) to re-define $\log p/q$. The consequence of this is that TRE is only defined when $p << m << q$, whereas sDRE is defined for any $m$ such that, $p << m$ and $q << m$. This implies that only under the following two conditions TRE and sDRE *telescopings* are equivalent: (1) $K = 1$ and (2) $m$ is such that $p << m << q$ and $q << m$. This condition holds when $p, q$ and $m$ have full support. However this condition easily breaks if, for example, $p$ and $q$ are mixtures of finite support distributions except for the trivial case when support of $m$ is exactly equal to the support of $q$. We demonstrate this in Figure 6 in Appendix B with a concrete example.

Another important and practical implication of this distinction is that, since $m$ only appears in the denominator in sDRE, it can also be used to improve model-free approaches to density ratio estimation, such as the fixed-design estimators based on the maximum mean discrepancy measure (Sugiyama et al., 2012). We show this in Section 5.3. It is also worth noting that unlike TRE, sDRE is realized as a single multi-class logistic regression that is trained using a softmax cross-entropy loss. This allows us to use the results from Ma & Collins (2018) who formally studied the distinction in the convergence of softmax cross-entropy and binary cross-entropy based NCE methods (e.g. sDRE vs TRE/BC-DRE). They showed that ranking-based noise contrastive estimation (Gutmann & Hyvärinen, 2010) (multi-class objective) is consistent under a much weaker assumption compared to the binary version. Specifically, in contrast to the binary version, the ranking based approach does not require the capacity to model the normalization constant of the distributions.

## 5 EXPERIMENTS

In this section, we provide a thorough empirical analysis on the accuracy and robustness of sDRE and demonstrate that sDRE outperforms all other baselines – BC-DRE, TRE (Rhodes et al., 2020), and F-DRE (Choi et al., 2021). We start with a toy 1D Gaussian dataset and fully analyze the behavior of sDRE. Then, we extend to high-dimensional complex tasks ranging from mutual information (MI) estimation to representation learning and show significant improvements on all accounts. Finally, we also show how sDRE can be applied to improve model-free DREs.

| $p$ | $q$ | GT-KL | BC-DRE | TRE | F-DRE | SDRE |
|---|---|---|---|---|---|---|
| $\mathcal{N}(0, 1 \times 10^{-6})$ | $\mathcal{N}(0, 1)$ | 13.32 | 1.82 | 2.92e-5 | 4.57 | 13.32 |
| $\mathcal{N}(-5, 1)$ | $\mathcal{N}(5, 1)$ | 50 | 69.56 | 34.51 | 15.96 | 53.97 |
| $\mathcal{N}(-1, 0.08)$ | $\mathcal{N}(2, 0.15)$ | 200.27 | 19.2 | 142.64 | 15.84 | 201.32 |
| $\mathcal{N}(-2, 0.08)$ | $\mathcal{N}(2, 0.15)$ | 355.82 | 17.95 | 227.98 | 14.49 | 358.81 |

Table 1: 1D density ratio estimation task. GT-KL stands for ground-truth KL Divergence.

## 5.1 1D EXPERIMENT

In the following 1D experiments, we use two 1D Gaussian distributions and consider the four setups described in Table 1. The first and second configurations (row-wise) are designed such that the distributions are only separated by HOD and FOD, respectively. The other two configurations represent the cases in which the distributions are separated by both FOD and HOD. In sDRE, we use the quadratic form $p(x) = wx^2 + b$ to model each of the densities and use Cauchy centered at 0 and linear mixing to construct $m$. We use KL-divergence estimation (expectation of the $\log$ ratios on samples from $p$) as the evaluation metric. We provide the exact setup for sDRE in Table 3 in Appendix C.

In all of the configurations, while the baseline methods, including TRE and F-DRE, fail to correctly estimate the KL divergences by a significant margin, sDRE reliably estimates them as shown in Table 1. Figure 7 in Appendix C illustrates that sDRE accurately estimates the underlying density ratios on the entire support of $m$ as well. These results demonstrate that it is better to use a single scaling measure whose samples overlaps with those from both $p$ and $q$, instead of using a chain of upto $K = 28$ *closely-packed* waymark distributions (see Table 3 in Appendix C for the exact TRE configurations). To provide further clarity into sDRE's density ratio estimation behavior, we analyze the reliability of its log ratio estimates using the Bayesian setup in Appendix D. The high accuracy of sDRE's KL divergence estimates can be attributed to the fact that the sDRE estimator is most certain and precise around the support of $p$, and the KL divergence estimates are computed with samples from $p$. This is illustrated in Figure 8 of Appendix D.

## 5.2 HIGH DIMENSIONAL EXPERIMENT

Following from TRE, we use the MI estimation problem from Belghazi et al. (2018); Poole et al. (2019) to evaluate sDRE on the more challenging higher-dimensional setup. In this problem, we estimate the mutual information, between a standard normal distribution and a Gaussian random variable $x \in \mathbb{R}^{2d}$ with a block-diagonal covariance matrix, where each block is $2 \times 2$ with 1 on the diagonal and $\rho$ on the off-diagonal. $\rho$ is computed given the number of dimensions and target mutual information with $I = -\frac{d}{2} \log(-\rho^2)$. Because this method of distribution construction only creates separation in HOD, we further adapt it by moving the means of the two distributions and create harder problems with both FOD and HOD. In sDRE, we model the densities of $p$, $q$, and the scaling measures $m_k$ with a quadratic form $p(x) = x^T W x + g_\phi(x) + b$, where $g_\phi$ is a linear layer. We use linear-mixing to construct $m_k$, where $K = 3$ or $K = 5$. We provide the exact configurations for sDRE in Table 4 in Appendix E. We describe the setups and report the results averaged across 3 runs with different random seeds in Table 2. In all of the settings, sDRE outperforms all other baselines, confirming the generalization of the results from the toy 1D experiments. These results corroborate our conjecture that TRE only solves problems with HOD. It is worth noting that, using only upto 5 scaling measures that are constructed using the linear mixing scheme, sDRE beats TRE substantially on cases with both FOD and HOD even though, TRE uses upto 15 waymarks that are also constructed using their linear mixing approach. This demonstrates, as per Lemma 1, our proposal of using the multi-class logistic regression does, in fact, prevent distribution shifts issues present in TRE when both FOD and HOD are present. F-DRE, in contrast, is only accurate in problems with FOD, but tends to underestimate the log ratios even more than BC-DRE on settings with MI greater than 20. Further details about this experiment such as how we chose $m$ and $K$ can be found in Appendix E, where we also provide plots of estimated log ratio vs the ground truth log ratio and training curves.

| Dim | $\mu_1, \mu_2$ | GT-MI | Single BC-DRE | TRE | F-DRE | sDRE |
|---|---|---|---|---|---|---|
| 40 | **0, 0** | **20** | $10.90 \pm 0.04$ | $14.52 \pm 2.07$ | $14.87 \pm 0.33$ | $18.81 \pm 0.15$ |
|  | **-1, 1** | **100** | $29.03 \pm 0.09$ | $33.95 \pm 0.14$ | $13.86 \pm 0.26$ | $119.96 \pm 0.94$ |
| 160 | **0, 0** | **40** | $21.47 \pm 2.62$ | $34.09 \pm 0.21$ | $12.89 \pm 0.87$ | $38.71 \pm 0.73$ |
|  | **-0.5, 0.6** | **136** | $24.88 \pm 8.93$ | $69.27 \pm 0.24$ | $13.74 \pm 0.13$ | $133.64 \pm 3.70$ |
| 320 | **0, 0** | **80** | $23.47 \pm 9.64$ | $72.85 \pm 3.93$ | $9.17 \pm 0.60$ | $87.76 \pm 0.77$ |
|  | **-0.5, 0.5** | **240** | $24.86 \pm 4.07$ | $100.18 \pm 0.29$ | $10.53 \pm 0.03$ | $217.14 \pm 6.02$ |

Table 2: High-dimensional mutual information estimation task.

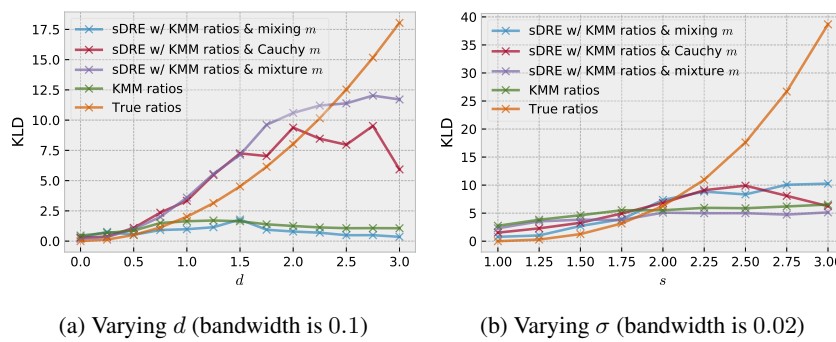

(a) Varying $d$ (bandwidth is 0.1)   (b) Varying $\sigma$ (bandwidth is 0.02)

Figure 3: Improving KL estimation between two standard Normals via KMM-DRE using sDRE

**Robustness and Generalization beyond Gaussian Distribution:**   We rigorously test the robustness of sDRE by evaluating it on the following setups. (1) $p$ and $q$ as 1D truncated normal distributions to evaluate it on another setup with distributions with finite support. (2) Randomized mean parameters to ensure that the sDRE model does not benefit from the symmetry of the distributions around 0 in Table 2. (3) $p$ and $q$ as Student-T distributions with randomized means to test the performance of sDRE under model mismatch. In all these settings, sDRE model is able to reliably estimate the true KL-divergence demonstrating that it is highly robust. Please see Table 5 in Appendix G for results and additional details.

## 5.3   IMPROVING KMM-BASED DRE WITH sDRE

We demonstrate the applicability of sDRE in the fixed design setup by applying it to the Kernel Mean Matching-DRE method (KMM-DRE; Huang et al., 2006; Sugiyama et al., 2012) for KL estimation. In *fixed design* setup, the DRE method directly outputs the densities for a set of samples without building a model. We consider two KL estimation experiments with increasing difficulty: (a) We estimate KL between $N(-d, 1)$ and $N(+d, 1)$ by varying $d$ from 0 to 3 and (b) We estimate KL between $N(0, s^2)$ and $N(0, 1/s^2)$ by varying $s$ from 1 to 3. For each experiment, we report the following KL estimates: (1) KL by MC using true ratios, (2) KL by MC using KMM-DRE estimates, and (3) KL by MC using estimates from KMM-DRE with sDRE for three types of $m$ construction: linear mixing, mixture, and Cauchy. We conduct the same experiments for different bandwidth $\sigma$, the parameter of the Gaussian kernel used by the KMM estimator. As it can be seen from Figure 14, sDRE consistently improves KMM-DRE in both experiments with different difficulty by improving the baseline KMM-DRE KL divergence estimates by up to 11 nats in some cases. We defer a detailed discussion of the results to Appendix H.

## 5.4   REPRESENTATION LEARNING IN SPATIALMULTIOMNIGLOT

Following the setup from Ozair et al. (2019); Rhodes et al. (2020), we apply sDRE to mutual information estimation and representation learning in the SpatialMultiOmniglot problem. The goal is to estimate the mutual information between $u$ and $v$, where $u$ is a $n \times n$ grid of Omniglot char-

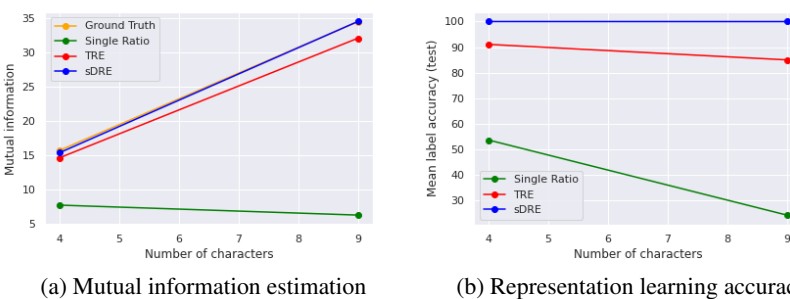

(a) Mutual information estimation       (b) Representation learning accuracy

Figure 4: SpatialMultiOmniglot representation learning results

acters from different Omniglot alphabets and $v$ is a $n \times n$ grid containing the next characters of the corresponding characters in $u$. After learning, we evaluate the representations from the encoder with a standard linear evaluation protocol (Oord et al., 2018). For the model, as in TRE, we use a separable architecture commonly used in MI-based representation learning literature and model the densities with $\log p(u, v) = g(u)^T W f(u)$, where $g$ and $f$ are 14-layer convolutional ResNets (He et al., 2015). We construct the scaling distributions via dimension-wise mixing. In this experiment, we only compare to the single ratio baseline and TRE because (Rhodes et al., 2020, Figure 4) already demonstrated that it significantly outperforms both Contrastive Predictive Coding (CPC) (Oord et al., 2018) and Wasserstein Predictive Coding (WPC) (Ozair et al., 2019) models of representation learning on the same exact task.

As can be seen in Figure 4a, sDRE outperforms both TRE and the single ratio baseline, exactly matching the ground truth MI. This improvement in MI estimation is reflected in the representations. Figure 4b illustrates that sDRE's encoder learns representations that achieve $\sim 100\%$ Omniglot character classification for both $d = n^2 = 4, 9$. On the other hand, the performances of the single ratio estimator and TRE (using the same exact dimension-wise mixing to construct waymark distributions) both degrade noticeably as the complexity of the task increases. As such, TRE only reach up to 91% and 85% respectively for $d = 4$ and $d = 9$. All models were trained with the same encoder architecture to ensure fair comparison. Please refer to Appendix F for further details and two additional experiments where we study the effects of varying $K$ and the encoder design.

## 6 CONCLUSION

In this work, we presented the alternative method of sDRE for density ratio estimation. We demonstrated that it has better theoretical grounding and improves the state-of-the-art in more challenging, high-dimensional density ratio estimation problems that have both FOD and HOD type of discrepancies. However, the presented framework has its limitations. For example, while sDRE estimates the ratio fairly well, (1) we do not provide any bounds on the estimation and (2) $m$ is not learned but provided to the model. We hope to address these issues in future work.

Moreover, our representation learning results suggest that improving MI estimation correlates with better representation learning. Therefore, another interesting direction for future work could be to explore if sDRE can improve MI-based representation learning methods, such as contrastive learning (e.g. Oord et al., 2018; Chen et al., 2020).

## 7 ETHICAL IMPACT

While density ratio estimation does not directly have major ethical concerns, it can, however, be used towards improving other deep learning methods of representation learning, generative models, etc. These work tend to have significant ethical considerations. For example, improved generative models using sDRE could potentially be used to advance the deep-fake methods that may be used with malicious intent. However, we hope that other applications of our work, such as those of MI estimation in science domains, outweigh the drawback of its potential illicit usages.

## 8    REPRODUCIBILITY STATEMENT

We have taken efforts to clearly describe all the experimental setups in separate appendices, one for each of the experiments. For the baseline methods, we have used models provided by the original authors. Where possible we have used existing evaluation tasks and metrics to report the results. Finally, in order to ensure reproducibility of our own results, we are releasing all the code at the time of submission. While most of our code is setup as Jupyter notebooks for easy accessibility, we will work to add documentation to improve the readability further.

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

## A    APPENDIX

### CONSTRUCTING $M$

We now elaborate on the three types of scaling measures that we used in this work.

**Overlapping Distribution:**    sDRE estimator, $\log \frac{p}{q} = \log \frac{p}{m} - \log \frac{m}{q}$ is defined when $p << m$ and $q << m$. Therefore, $m$ needs to such that its support contains the supports of $p$ and $q$. Any distribution with full support such as the normal distribution trivially satisfies this requirement. However, satisfying this requirement does not guarantee empirical overlap of the distributions $p$, $q$ with $m$ in finite sample setting. In order to ensure overlap of samples between the two pairs of distributions we recommend the following:

- Heavy-tailed Distributions: Distributions such as, Cauchy and Student-t are better choice for $M$ compared to the normal distribution. This is because their heavier tails allow for easily connecting $p$ and $q$ with higher sample overlap when they are far apart (especially in the case of FOD).

- Mixtures: Another way to connect $p$ and $q$ using $m$ such that they have their samples overlap, is to use the mixture distribution. Here, we first convolve $p$ and $q$ with a standard normal and then take equal mixtures of the two.

- Truncated Normal: If $p$ and $q$ have finite support, one can also use a truncated normal distribution or a uniform distribution that at least spans over the entire support of $q$. This is assuming that $p << q$.

**Linear Mixing:**    In this construction scheme, distribution $M$ is defined as the empirical distribu-tion of the samples constructed by linearly combining samples $X_p = \{x_p^i\}_{i=1}^N$ and $X_q = \{x_q^i\}_{i=1}^N$ from distributions $p$ and $q$ respectively. That is, $m$ is the empirical distribution over the set $X_m = \{x_m^i | x_m^i = \alpha x_p^i + (1 - \alpha)x_q^i, x_p \in X_p, x_q^i \in X_q\}_{i=1}^N$. This construction is related to the linear combination waymark of Rhodes et al. (2020). In TRE the waymark distribution is defined as the empirical distribution of the set $X_m = \{x_m^i | x_m^i = \sqrt{1 - \alpha^2}x_p^i + \alpha x_q^i, x_p \in X_p, x_q^i \in X_q\}_{i=1}^N$.

This weighting scheme skews the samples from the waymark distribution towards $p$. Therefore, care needs to be taken when $p$ and $q$ are finite support distributions so that the samples from the waymark distributions do not fall out of the support of $q$.

Using either of the weighting schemes, one can construct $K$ different scaling measures. sDRE can either use these $K$ scaling measures separately using a K+2-way classifier or define a single mixture distribution using them as component distributions and train a 3-way classifier. We refer to this construction as Mixture of Linear Mixing.

**Dimension-wise Mixing:**    In this construction scheme, that is borrowed from TRE as it is, $M$ is defined as the empirical distribution of the samples generated by combining different subsets of dimensions from samples from $p$ and $q$. We describe the exact construction scheme from TRE below for completeness:

Given a $d$-length vector $x$ and that $d$ is divisible by $l$, we can write down $x = (x[1], ...x[l])$, where each $x[i]$ has the length $d/l$. Then, a sample from the $k$th scaling measure is given by: $x_k^i = (x_q^i[1], ...x_q^i[j], x_p^i[j+1], ..., x_p^i[l])$, for $j = 1, ..., l$), where $x_p^i \sim p$ and $x_q^i \sim q$ are randomly paired.

# B    APPENDIX

## TRE VS sDRE

In Figure 2b in Section 3, we showed that in comparison to TRE that uses 3 intermediate waymark distributions, sDRE is able to significantly better estimate the ground truth density ratio using just a single Cauchy scaling measure that we introduced. To demonstrate that this improved performance does not only stem from the novel $m$, but also from the use of a single multi-class logisitic regression (afforded by the exact form of sDRE telescoping), we train sDRE using TRE's waymark distributions as scaling measures. This is illustrated in Figure 5c. Unlike TRE that does not estimate $\log \frac{p}{q}$ accurately on samples from $p, q, p_1$ and $p_3$, sDRE is able to estimate $\log \frac{p}{q}$ on all 5 distributions significantly better even with sub-optimal scaling measures. The impact of using the sub-optimal scaling measure in sDRE on the quality of density ratio estimation can be teased out by comparing to the results in Figures 5b where sDRE is trained using Cauchy for $m$.

**Finite Support Distribution Example for $K = 1$:**    For the case, $K = 1$, in Section 4.3 we theoretically showed that TRE and sDRE are not equivalent in general. We now demonstrate this with a specific example in Figure 6. Here we set $p = 0.5 \times \mathcal{TN}(-1, 0.1, low = -1.1, high = -0.9) + 0.5 \times \mathcal{TN}(1, 0.1, low = 0.9, high = 1.1)$ and $q = 0.5 \times \mathcal{TN}(-1, 0.2, low = -1.2, high = 0.8) + 0.5 \times \mathcal{TN}(1, 0.2, low = 0.8, high = 1)$, as shown in Figure 6a. We set both, the waymark distribution TRE and the scaling measure for sDRE to $m = \mathcal{TN}(0, 1, low = -1.2, high = 1.2)$ using the proposed *overlapping distribution* construction. As such, $p << q << m$ and therefore, TRE is undefined for the second term as $\frac{m}{q}$ is not defined for samples from $m$ that are outside the support of $q$. On the other hand, sDRE is well defined as $\frac{q}{m}$ is finite and defined over the entire support of $m$. It can be clearly seen in Figure 6b that TRE estimator for $\frac{m}{q}$ blows up to very high values on samples from $m$ where $q$ does not have any support. This, however, is not the case with sDRE, which accurately estimates $\frac{q}{m}$ as 0 on samples outside the support of $q$ as shown in Figure 6c.

Despite being undefined, when used with our proposed $m$, TRE may still estimate $\log r_{p/q}$ accurately on samples from $p$. We conjecture that this is because, the numerical estimation of the $dM/dQ$ is finite over the support of $p$. As such, care needs to be taken when equating TRE to sDRE to ensure that the relevant RNDs are well defined along with of the accuracy of the numerical estimation.

(a) TRE on $p = \mathcal{N}(-1, 0.1)$ and $q = \mathcal{N}(1, 0.2)$ with TRE's skewed linear waymarks

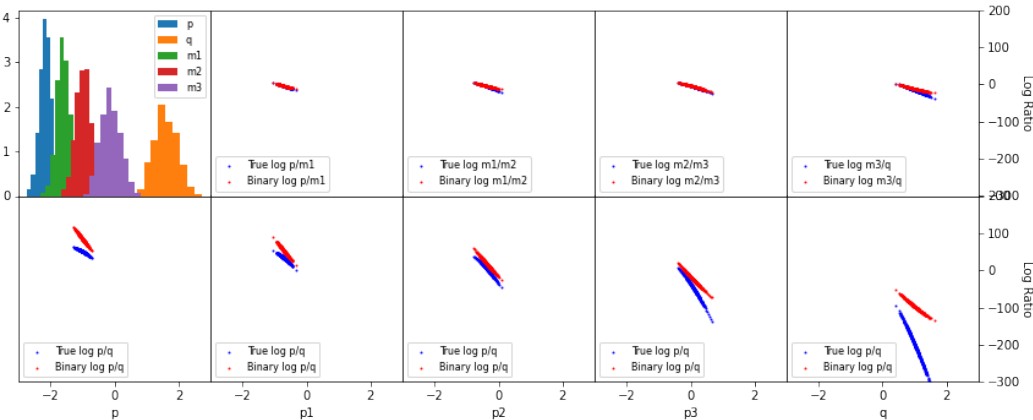

(b) sDRE on $p = \mathcal{N}(-1, 0.1)$ and $q = \mathcal{N}(1, 0.2)$ using $m = \text{Cauchy}(0, 1)$ as the scaling measure.

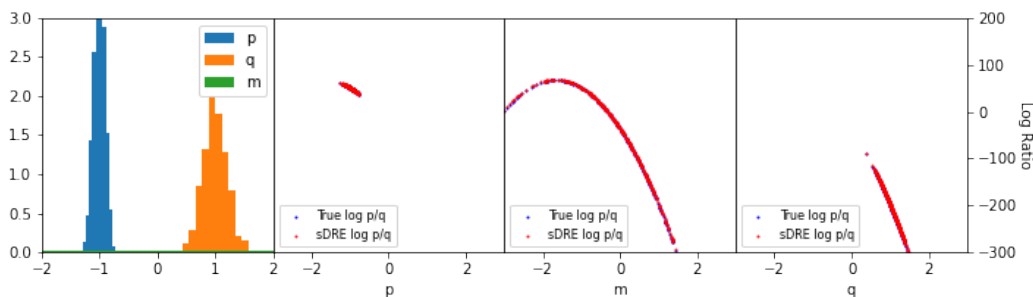

(c) sDRE using TRE waymarks as scaling measures.

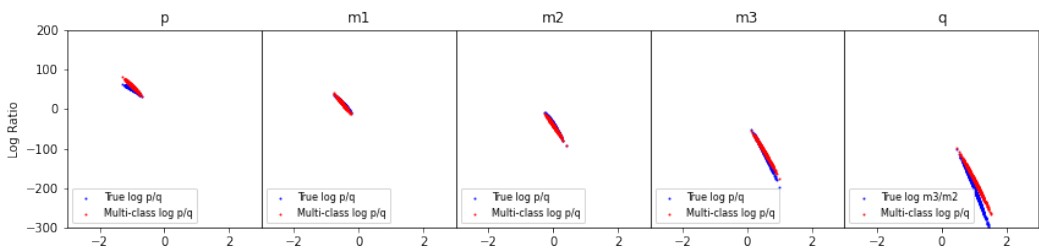

Figure 5: TRE vs sDRE on $p = \mathcal{N}(-1, 0.1)$ and $q = \mathcal{N}(1, 0.2)$ using 3 intermediate distributions $p_1, p_2, p_3$ constructed using the *linear-combination* construction.

(a) Setup        (b) TRE: $dM/dQ$        (c) sDRE: $dQ/dM$

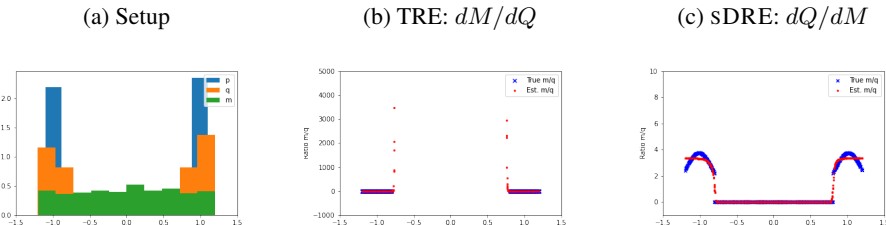

Figure 6: sDRE vs TRE telescoping on mixtures of distributions with finite support

| Figure Label | $p$ | $q$ | TRE $p_k$ | sDRE $m$ |
|---|---|---|---|---|
| a | $\mathcal{N}(0, 1e-6)$ | $\mathcal{N}(0,1)$ | Linear Mixing with $\alpha$ = [6.10e-05, 0.0078, 0.13] | Mixture of Linear Mixing with $\alpha$ = [6.10e-05, 0.0078, 0.13] |
| b | $\mathcal{N}(-1, 0.08)$ | $\mathcal{N}(2, 0.15)$ | Linear Mixing with $\alpha$ = [0.053, 0.11, 0.16, 0.21, 0.26, 0.31, 0.37, 0.42, 0.47, 0.53, 0.58, 0.63, 0.68, 0.74, 0.79, 0.84, 0.89, 0.95] | $\mathcal{C}(0,1)$ |
| c | $\mathcal{N}(-2, 0.08)$ | $\mathcal{N}(2, 0.15)$ | Linear Mixing with $\alpha$ = [0.03, 0.07, 0.1, 0.14, 0.17, 0.21, 0.24, 0.28, 0.31, 0.34, 0.38, 0.41, 0.45, 0.48, 0.52, 0.55, 0.59, 0.62, 0.66, 0.69, 0.72, 0.76, 0.79, 0.83, 0.86, 0.9, 0.93, 0.97] | $\mathcal{C}(0,1)$ |
| d | $\mathcal{N}(-5, 1)$ | $\mathcal{N}(5, 1)$ | Linear Mixing with $\alpha$ = [0.11, 0.22, 0.33, 0.44, 0.55, 0.66, 0.77, 0.88] | $\mathcal{C}(0,2)$ |

Table 3: Experiment configurations for Figure 7

## C  APPENDIX

### 1D DENSITY RATIO ESTIMATION TASK

In Section 5.1 we used KL-divergence as the evaluation metric to assess the accuracy of the density ratio estimation of each of the models, BC-DRE, TRE, F-DRE and sDRE. Since KL-divergence only evaluate the density ratio over samples from $p$, here we provide the plots of the log density ratio for all the models over a larger interval to better capture their behavior.

The configurations of the experiments are described in Table 3. Column 1 (Figure Labels) corresponds to the respective columns in Figure 7. The rows of Figure 7 represent the four models in this order: BC-DRE, TRE, F-DRE, and sDRE. Except for the first configuration (Figure Label a), which was trained with 1,000 samples in total, all other configurations are trained with 100,000 samples in total for all four models.

As evident from the figure, sDRE is the only model that not only accurately estimates the KL divergences, but also perfectly estimates the log density ratios on the entire support of $m$.

## D  APPENDIX

### UNCERTAINTY QUANTIFICATION OF SDRE LOG-RATIO ESTIMATES WITH HAMILTONIAN MONTE CARLO

In the 1D experiments, sDRE consistently led to highly accurate KL divergence estimates even in challenging settings where state-of-the-art methods fail. To understand why sDRE gives such accurate KL estimates, we conduct an analysis on the reliability of its log-ratio estimates by analyzing the distribution of the estimates in a Bayesian setup, and study how it impacts the KL divergence estimation. For this analysis, we use a classifier with the standard normal distribution as the prior on its parameters. The distribution of the log-ratio estimates is simply the distribution of the estimates from the classifiers with different posterior parameters, which are sampled. We consider the setup where $p = \mathcal{N}(-1.0, 0.1)$ and $q = \mathcal{N}(1.0, 0.2)$ and draw samples from the posterior using an Hamil-

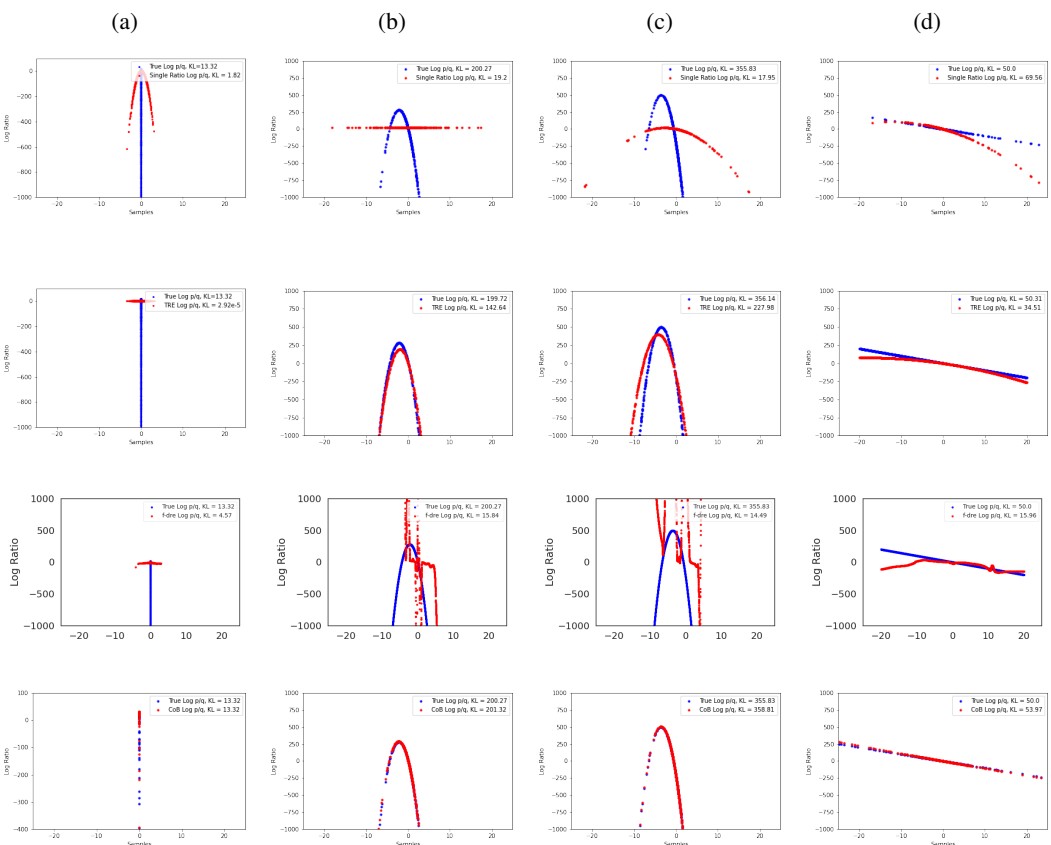

Figure 7: 1D density ratio estimation task. The configurations of the 1D Gaussians are the following in column-wise order: (a) $p = \mathcal{N}(0, 1e-6), q = \mathcal{N}(0, 1)$, KL= 13.32 (b) $p = \mathcal{N}(-1, 0.08), q = \mathcal{N}(2, 0.15)$, KL = 200.27 (c) $p = \mathcal{N}(-2, 0.08), q = \mathcal{N}(2, 0.15)$, KL = 355.82 (d) $p = \mathcal{N}(-5, 1), q = \mathcal{N}(5, 1)$, KL = 50.0. The models are the following row-wise: single ratio BC-DRE, TRE, F-DRE, and sDRE.

tonian Monte Carlo (HMC) sampler. We then compute a set of samples of the log-ratio estimates from sDRE and estimate the mean and quantiles using these samples. Figure 8 shows these results. We find that sDRE is accurate and manifests lowest uncertainty around the mean $(-1.0)$ of $p$. The uncertainty increases as we move away from the modes of distributions $p$ and $q$. Since KL divergence is the expectation of the log-ratio on samples from $p$ and the high density region of $p$ exactly matches the high confidence region of sDRE, it is able to consistently estimate the KL divergence accurately even when $p$ and $q$ are far apart.

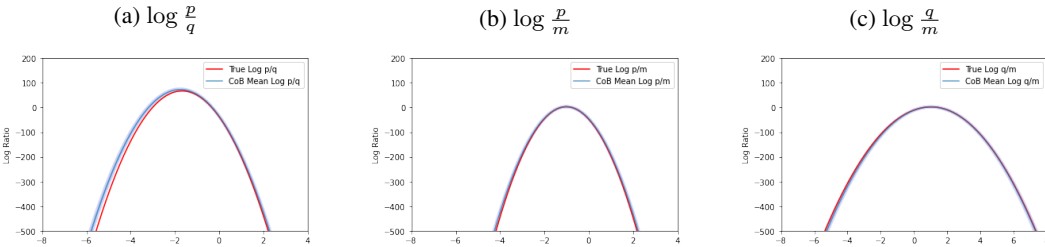

Figure 8: Uncertainty quantification for sDRE estimator. Here, $p = \mathcal{N}(-1.0, 0.1), q = \mathcal{N}(1.0, 0.2)$ and $m = \mathcal{C}(0, 1)$. We plot the 3x standard deviation around the mean in light blue.

| Dim | MI | $m$ **using LM** |
|-----|-----|-----------------|
| 40  | 20  | [0.25,0.5,.75] |
|     | 100 | [0.35,0.5,.85] |
| 160 | 40  | [0.25,0.5,.75] |
|     | 136 | [0.15,0.35,0.5,.75,.95] |
| 320 | 80  | [0.25,0.5,.75] |
|     | 240 | [0.15,0.35,0.5,.75,.95] |

Table 4: Configuration of SDRE for the high dimensional experiments. LM stands for Linear Mixing

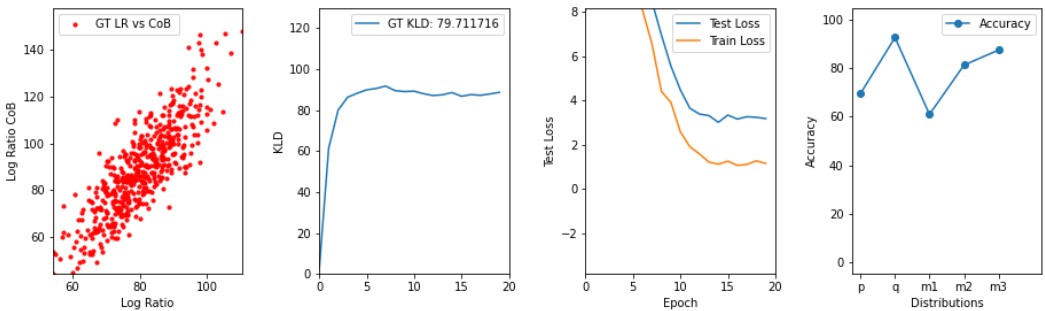

Figure 9: Diagnostic plot for a high dimensional experiment.

# E  APPENDIX

## HIGH DIMENSIONAL EXPERIMENT

In Section 5.2, we showed that SDRE performs better than all baseline models when $p$ and $q$ are high dimensional Gaussian distributions. Prior work of Rhodes et al. (2020) has considered high dimensional cases with HOD only, whereas here we additionally consider cases with FOD and HOD to provide a more complete picture. Our results show that SDRE outperforms all other methods on the task of MI estimation as the function of the estimated density ratio. It is worth noting that SDRE uses only upto 5 scaling measures that are constructed using the linear mixing scheme and beats TRE substantially on cases with both FOD and HOD, although TRE uses upto 15 waymarks also constructed using linear mixing approach. This demonstrates, as per Lemma 1, our proposal of using the multi-class logistic regression does, in fact, prevent distribution shifts issues of TRE when both FOD and HOD are present and help estimate the density ratio more accurately.

We now describe the SDRE configuration and other setup related details.

**Scaling Measures:**  For all the high dimensional experiments throughout this work, we construct $m$ using the linear mixing scheme as described in Appendix A. Table 4 provides the number $K$ of scaling measures along with the exact mixing weights for each of the 6 settings.

As a general principle, we chose these three sets of mixing weights so that their cumulative samples overlap with the samples of $p$ and $q$ similar to how the heavy tailed distribution worked in the 1D case. Please note, unlike the 1D experiment, the HOD in these experiments is significantly higher. Therefore, using a heavy tailed distribution does not work. Linear mixing $p$ and $q$ on the other hand, mixes first and higher order statistics and therefore, populates samples that overlap with both $p$ and $q$. In some cases, we found that a mixture of linear mixing with $K = 1$ can also be used to estimate the density ratio. However, this requires using a neural network-based classifier and requires much more tuning of the hyperparameters.

For choosing $K$, we use a grid search based approach. We monitor the classification accuracy across all the $K + 2$ distribution. If this accuracy is very high ($> 95\%$ for all classes), this implies that the classification task is easy and therefore the DRE may suffer from the density chasm issue. On

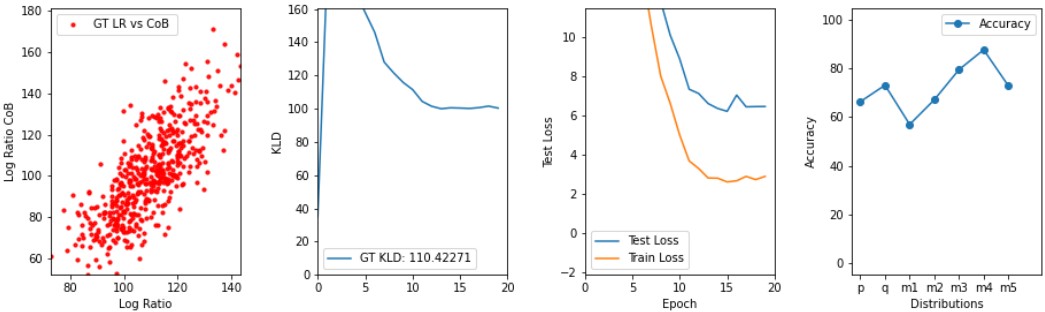

Figure 10: Diagnostic plot for a high dimensional experiment with randomized means.

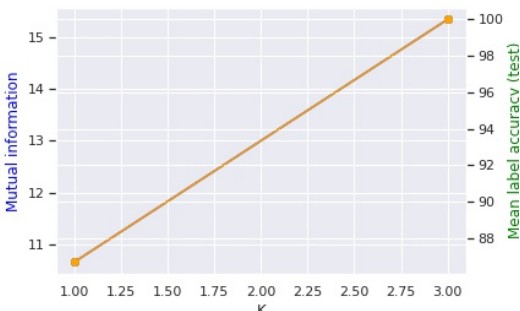

Figure 11: Mutual information estimation and representation learning with varying K.

the other hand, if the classification accuracy is too low ($< 50\%$ for all classes), then again, the DRE does not estimate well. We found that targeting an accuracy curve as shown in Figure 9 (last panel) leads to optimal density ratio estimation. This curve plots the test accuracy across all the classes and, empirically when it stays between the low and the high bounds of (50%,95%), the DRE estimates the ratios fairly well. The first panel shows that sDRE estimates the ground truth ratio accurately across samples from all the $K + 2$ distributions, the second panel shows that KL estimates of sDRE is close to the ground truth KL and the third panel shows that both test and training losses have converged. Figure 10 shows another example for the case of randomized means. While sDRE also manages to get the ground truth KL correctly and most of the ratio estimates are also accurate, it does, however, slightly overestimate the log ratio for some of the samples from $p$.

**Additional Baseline Comparisons** We also compare to Kato & Teshima (2021) in a high-dimensional setting (row 2 of 2 with dimensionality of 40 and $\mu_1 = -1, \mu_2 = 1$. In this setting in which the ground truth MI is 100, while sDRE meaningfully estimates the MI as 119.96, the best model from Kato & Teshima (2021) only estimates it to be 1.60.

# F APPENDIX

## SPATIALMULTIOMNIGLOT EXPERIMENT

SpatialMultiOmniglot is a dataset of paired images $u$ and $v$, where $u$ is a $n \times n$ grid of Omniglot characters from different Omniglot alphabets and $v$ is a $n \times n$ grid containing the next characters of the corresponding characters in $u$. In this setup, we treat each grid of $n \times n$ as a collection of $n^2$ categorical random variables, not the individual pixels. The mutual information $I(u, v)$ can be computed as: $I(u, v) = \sum_{i=1}^{n^2} \log l_i$, where $l_i$ is the alphabet size for the $i^{th}$ character in $u$. This problem allows us to easily control the complexity of the task since increasing $n$ increases the mutual information.

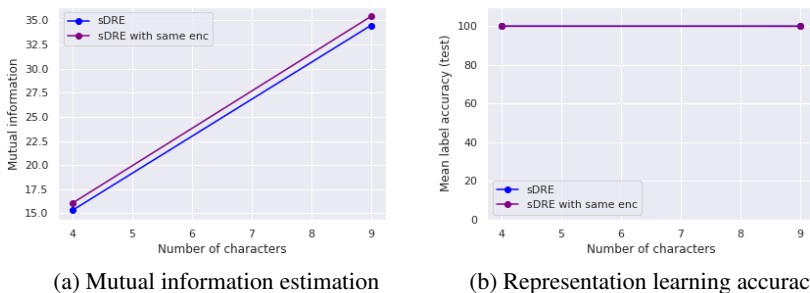

(a) Mutual information estimation    (b) Representation learning accuracy

Figure 12: SpatialMultiOmniglot representation learning results with same encoder for $f$ and $g$.

For the model, as in TRE, we use a separable architecture commonly used in MI-based representation learning literature and model the densities with $\log p(u, v) = g(u)^T W f(u)$, where $g$ and $f$ are 14-layer convolutional ResNets (He et al., 2015). We construct the scaling distributions via dimension-wise mixing – exactly the way that TRE does.

To evaluate the representations after learning, we adopt a standard linear evaluation protocol to train a linear classifier on the output of the frozen encoder $g(u)$ to predict the alphabetic index of each character in the grid $u$.

ADDITIONAL EXPERIMENTS

In addition to the experiments in the main text, we run two more experiments with SpatialMultiOmniglot to test the effect of the size of $K$ and the effect of using the same encoder for $g$ and $f$ (i.e, modeling the densities with the form $\log p(u, v) = g(u)^T W g(v)$).

**Varying $K$:** We test the effect of changing $K$ in the $d = 4$ experiment. For $K = 1$, we aggregate all the dimension-wise mixed samples into 1 class, whereas for $K = 3$, we separate them into their respective classes (corresponding to the number of dimensions mixed). We illustrate this effect in 11. In line with the finding of Ma & Collins (2018), increasing the number of K not only helps the model reach the ground truth MI, but also the quality of representations worsens from $100\%$ to $86.7\%$ test classification accuracy.

**Single Encoder Design:** Furthermore, we test the contribution of using two different encoders $f$ and $g$ instead of one. As seen in Figure 12, in both cases of $d = 4, 9$, the two models reach slightly different but similar MI estimates, but, interestingly, do not differ at all in the test classification accuracy. Empirically, we also found that using one encoder helps the model converge to much faster. Overall, this experiment demonstrates that using two different encoders does not necessarily work to our advantage.

# G APPENDIX

ROBUSTNESS AND MODEL MISMATCH ANALYSIS

In order to evaluate the robustness of SDRE method, we use the following setups.

**Breaking Symmetry:** In the experiment setup for the high dimensional experiments as described in Section 5.2, the means of the Gaussian distribution $p$ and $q$ were symmetrical in majority of the cases. In order to ensure that this symmetry around zero did not provide an advantage to the sDRE method, we also evaluate it on Gaussian $p$ and $q$ with randomized means for the two settings shown in rows 2 and 3 of Table 5. Beside the mean parameters, we do not change any other configuration in the experiment or the model. sDRE manages to estimate the ground truth KL divergence accurately in this case, therefore establishing that it did not benefit unfairly from the symmetry of distributions around zero in the high dimensional experiments.

**Model Mismatch:** In rows 3, 5 and 6 of Table 5, we evaluate SDRE by replacing one or both distributions $p$ and $q$ from Gaussian in the case above with Student-t distribution of the same scale with randomized means. We set the degrees of freedom to 10. These experiments test how SDRE performs when there is a model mismatch, i.e. how SDRE performs using the same quadratic model that was used when $p$ and $q$ were set to be Gaussian with lighter tails. As can be seen, SDRE seems to be very robust in these cases and is able to accurately estimate the ground truth KL.

$p$ **and** $q$ **with Finite Support:** Finally, we test SDRE on another case where $p$ and $q$ are finite support distributions that have both FOD and HOD. For this, we set them to truncated normal, as shown in row 1. This setting is similar to the 1D example with Gaussian $p$ and $q$ that we have used throughout the manuscript. We also set $m$ to be truncated normal with scale set to 2 in order to allow it to have overlap with both $p$ and $q$. As expected, SDRE not only manages to correctly estimate the ground truth KL, but also correctly decays its log ratio estimates when the samples do not fall under the support of $p$ as shown in Figure 13.

| Dim | $p$ | $q$ | $m$ | **GT-KL** | **SDRE** |
|---|---|---|---|---|---|
| **1** | Truncated Normal loc=-1, scale=0.1 (-1.1,-0.9) | Truncated Normal loc=1, scale=0.2 (-1.1,1.2) | Truncated Normal loc=-1, scale=2 (-1.1,1.2) | 50.65 | 52.35 |
| **160** | Normal loc=R(-.5,.5), cov=$2 \times 2$ BD | Normal loc=R(-.5,.5), cov=$I$ | LM | 54.29 | 54.10 |
| **160** | Normal loc=-1, cov=$2 \times 2$ BD | Mixture of Normal loc=0.9, cov=$I$ and Normal loc=0.9, cov=$I$ | LM | 105.60 | 98.27 |
| **320** | Student T loc=R(-.5,.5), scale=$2 \times 2$ BD, df=10 | Student T loc=R(-.5,.5), scale=$I$, df=10 | LM | 53.82 | 51.03 |
| **320** | Normal loc=R(-1,1), cov=$2 \times 2$ BD | Normal loc=R(-1,1), cov=$I$ | LM | 110.05 | 102.63 |
| **320** | Student T loc=R(-1,1), scale=$2 \times 2$ BD, df=10 | Student T loc=R(-1,1), scale=$I$, df=10 | LM | 103.12 | 113.53 |
| **320** | Normal loc=0, cov=$2 \times 2$ BD | Student T loc=0, scale=$I$, df=20 | LM | 82.02 | 83.63 |

Table 5: Robustness evaluation for SDRE. Here R(a,b) stands for randomized mean vector where each dimension is sampled uniformly from the interval $(a, b)$. LM stands for the linear mixing construction scheme.

# H    APPENDIX

## IMPROVING KMM-BASED DRE WITH SDRE

Another advantage of SDRE is that it is applicable to DRE methods with a *fixed design* setup, i.e. a DRE method that directly outputs the densities for a set of samples without building a model; TRE is not applicable to DRE methods in this setup as it models the ratios directly. The flexibility of our model derives from the fact $m$ is in the denominator of both ratios. An example of DRE methods with a fixed design setup is the kernel mean matching (KMM) based DRE method (KMM-DRE;

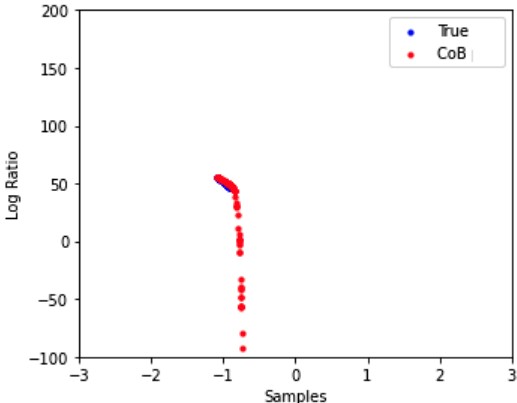

Figure 13: sDRE is able to match the ground truth log-ratio and correctly decays all the values towards $-\infty$ if they don't fall under the support of $p$.

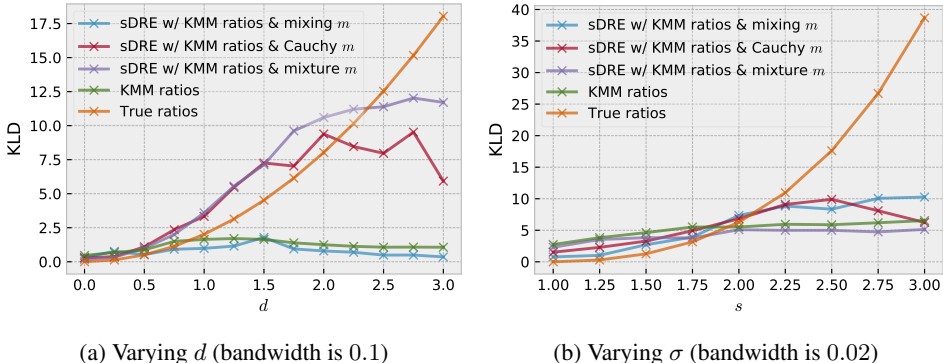

(a) Varying $d$ (bandwidth is 0.1)      (b) Varying $\sigma$ (bandwidth is 0.02)

Figure 14: Improving KL estimation between two standard Normals via KMM-DRE using sDRE

Huang et al., 2006; Sugiyama et al., 2012). The main advantage of this method is that the ratio estimators can be computed in closed form and therefore are computationally more efficient.

We demonstrate the applicability of sDRE in the fixed design setup by applying sDRE to the KMM-DRE method for KL estimation. In order to show the benefit of sDRE, we consider two KL estimation experiments with increasing difficulty: (a) We estimate KL between $N(-d, 1)$ and $N(+d, 1)$ by varying $d$ from 0 to 3 and (b) We estimate KL between $N(0, s^2)$ and $N(0, 1/s^2)$ by varying $s$ from 1 to 3. For each experiment, we report the following KL estimates: (1) KL by MC using true ratios, (2) KL by MC using KMM-DRE estimates, and (3) KL by MC using estimates from KMM-DRE with sDRE for three types of $m$ construction: linear mixing, mixture, and Cauchy.[1] We conduct the same experiments for different bandwidth $\sigma$, the parameter of the Gaussian kernel used by the KMM estimator. Figure 14 summarizes the results. Overall, sDRE significantly improves the baseline KMM-DRE KL divergence estimates by up to 11 nats.

As it can be seen from the figures, sDRE consistently improve KMM-DRE in both experiments with different difficulty. In Figure 14a, it can be seen sDRE with mixture and Cauchy as $M$ gives KL estimates closer to the true value even when $d = 2.5$, while KMM-DRE fails even for $d = 1.0$. In Figure 14b, we can see sDRE with mixing $M$ helps calibrate the KL estimates from KMM-DRE: For $s \leq 1.75$, KMM-DRE *overestimates* the KL and sDRE gives estimates closer to the true KL by giving a *lower* estimate; for $s \geq 1.75$, KMM-DRE *underestimates* the KL and sDRE gives estimates closer to the true KL by giving a *higher* estimate. It is also worth noting that different

---

[1]For KMM-DRE, we solve the constrained optimization problem using the solvers from JuMP.jl, via DensityRatioEstimation.jl. We set the kernel bandwidth to 0.1 and unit sum tolerance to 0.

types of $M$ helps in the two experiments as experiment (a) is a case of FOD and (b) is a case of HOD.

