# OpenReview forum: "Scaling Densities For Improved Density Ratio Estimation"
_ICLR.cc/2022/Conference — ICLR 2022 Submitted_

### Official Review · Reviewer_RSpk · 2021-11-01

**Correctness:** 2
**Technical Novelty And Significance:** 2
**Empirical Novelty And Significance:** 3
**Recommendation:** 3
**Confidence:** 4

**Main Review:**

1. The descriptions and titles should be improved.
The paper does not seem to be written in a clear way. For instance, the title in 2.1 is logistic regression but the contents are general results. The title of 2.2 is density chasm but the authors have never explained what it is.
Also, while the authors kept mentioning 'logistic regression' in a couple of places (such as Sec 4.1), the results are not limited to logistic regression.
The logistic regression places a linear model of the log-odds (i.e., $\log \eta(x)$ in the paper), which is a restricted approach in some sense.
Also, there is a typo in Lemma 1 that $\hat r(x)$ should be $\frac{\hat \eta(x)}{1-\hat \eta(x)}$.
Moreover, the quantity $\pi_C$ in Lemma 1 was never defined.
In the first paragraph below Lemma 1, it stated that 'the reference density $C$'. But shouldn't $C$ be the index rather than a density?

2. Misleading statement: "This empirically confirms that, unlike TRE, the SDRE estimator is theoretically guaranteed ..." (Page 5).
This statement does not make any sense.
Theoretical guarantee is not from empirical analysis. It should be derived from symbolic analysis (e.g., mathematical derivation).


3. Lack of novelty?
It seems to me that the only novel part was to use a 'single' intermediate density to estimate the ratio and replace the $\log [m/q]$ by $-\log [q/m]$.
I don't think this is something that novel.
sDRE has considered a more general version (a sequence) of this idea (See Sec. 4.3). And replacing the log by -log is an elementary approach that many people did.
The paper has to provide more analysis and justifications on its contribution.


4. Choice of the measure M needs more analysis.
This part is the key to this paper since the paper is built on a single intermediate density.
This paper will be improved a lot if the authors can provide a detailed analysis and argument on how to choose a good measure $M$ and even prove some useful theorems (a better convergence rate, an elegant representation, an efficient algorithm).
However, the authors only mentioned three ideas without much analysis on each of them.


5. Estimation/Learning of $\eta(x)$.
It was never explicitly explained how the authors estimate the class-probability $\eta(x)$.
From the contents, I guess the authors apply a logistic regression method. But again, $\eta(x)$ can be estimated by other approach such as a tree classifier. Is the logistic regression offers any benefits on the later analysis? Or can we use other generative classifiers? Questions like this should be answered.


6. A bit of redundant of FOD and HOD.
It was not clear to me why the authors introduce the concept of FOD and HOD. First, these concepts were not well-defined beyond Gaussian. Second, these concepts were not related to the proposed method and were only used in the experiments. Why not just call them 'mean-shift' Gaussian problems and `variance-shift' Gaussian problems?



**Summary Of The Paper:**

This paper proposed to estimate the (log) density ratio, $\log [p/q]$ by introducing an intermediate density $m$, and rewrite it as $\log [p/q] = \log [p/m] - \log [q/m]$. The authors then estimate each ratio by a generative classifier (logistic regression). Some simulations were conducted to argue the benefit of the proposed method.

**Summary Of The Review:**

This paper lacks several important analysis on the proposed methods and the writing was misleading and confusing. The paper requires a lot of work to improve.

---

> ### Author Response · Authors · 2021-11-12
> **Response: Part 1**
>
> We thank the reviewer for their feedback. __We want to start by pointing out that the reviewer has largely misunderstood our method and therefore the provided summary is incorrect. We do not estimate each ratio by a generative classifier. In fact, our work proposes the opposite, i.e. estimating all the ratios using a single multi-class classifier.__ We now address each of their concerns one by one.
>
> ---
>
> > The descriptions and titles should be improved. The paper does not seem to be written in a clear way. For instance, the title in 2.1 is logistic regression but the contents are general results.
>
> It is true that Lemma 1 is general to any multi-class classifier and not limited to logistic regression only. We have amended the title as per reviewer’s suggestion.
>
> > Also, while the authors kept mentioning 'logistic regression' in a couple of places (such as Sec 4.1), the results are not limited to logistic regression.
>
> While Lemma 1 is general and applies to any multi-class classifier, this work presents a multi-class logistic regression based DRE. Hence, we refer to it throughout the paper.
>
> > The title of 2.2 is density chasm but the authors have never explained what it is.
>
> We state the density chasm problem in line 2 of section 2.2 starting with, “This phenomenon...”. Please note that there is no formal definition for density chasm problem at present so we provided a rephrased description from Rhodes et at. (2020).
> We have added a line to point out that this is the density chasm issue.
>
> > The logistic regression places a linear model of the log-odds (i.e., log⁡η(x) in the paper), which is a restricted approach in some sense.
>
> With logistic regression, we do not mean to imply “linear logistic regression”. The regression function in our work is nonlinear.
>
> >  Also, there is a typo in Lemma 1 that r^(x) should be η^(x)1−η^(x).
>
> Sorry about the typo. The correct notation is $n_C(x)$ in the denominator. We have changed it.
>
> > Moreover, the quantity πC in Lemma 1 was never defined. In the first paragraph below Lemma 1, it stated that 'the reference density C'. But shouldn't C be the index rather than a density?
>
> We apologize for the missing definition and have added it to the manuscript. $\pi_C$ is the prior class probability for class C. We have also changed the manuscript to ensure that we refer to $C$ as the class index and the corresponding density $p_C$ as the reference distribution of X indexed by C.
>
> > Misleading statement: "This empirically confirms that, unlike TRE, the SDRE estimator is theoretically guaranteed ..." (Page 5). This statement does not make any sense. Theoretical guarantee is not from empirical analysis. It should be derived from symbolic analysis (e.g., mathematical derivation).
>
> We apologize if the reviewer found our statement misleading. We have rephrased it. However, the quoted statement was intended to say that the empirical results are in consensus with the theoretical guarantee and are not provided in lieu of theoretical proof. More precisely, we first presented Lemma 1 that provides a theoretical guarantee that sDRE can estimate each of the density ratios without suffering from the distribution shift issue. Then, we show that empirically, it holds in our experiments.
>
> > Lack of novelty? It seems to me that the only novel part was to use a 'single' intermediate density to estimate the ratio and replace the log⁡[m/q] by −log⁡[q/m]. I don't think this is something that novel. sDRE has considered a more general version (a sequence) of this idea (See Sec. 4.3). And replacing the log by -log is an elementary approach that many people did. The paper has to provide more analysis and justifications on its contribution.
>
> We strongly disagree with the reviewer. There may be a misunderstanding of our main contribution. The novelty of sDRE is not in simply flipping the log or in the introduction of new M measures. It is two fold:
>
> * __Telescoping vs Scaling:__ sDRE takes a different approach to the telescoping than the one presented in Rhodes et al, (2020). Instead of computing a chain of K intermediate density ratios to estimate $p/q = p/m_1 \times m_1/m_2 \dots \times m_K/q$, sDRE proposes to scale the densities $p$ and $q$ by one of K different M measures. This allows for computing the log ratio as $\log p/q = \log p/m_i - \log q/m_i$ using any of the K M-measures instead of chaining them. This is explained in the second last paragraph of section 4.1.
>
> * __Multi-class vs Binary Classification:__ sDRE uses a multi-class classifier (with softmax cross entropy loss) as a DRE. As such, the log ratio between two distributions is represented by the difference of their corresponding logits (see section 4.1). This, using Lemma 1, is shown to not suffer from the distribution shift issues unlike the chain of binary  classifier based DRE and leads to a drastic improvement in density ratio estimation compared to the state-of-the-art.
>
> __Please see part 2 for the rest of the response__

---

> > ### Author Response · Authors · 2021-11-12
> > **Response: Part 2**
> >
> > __Continuing from Part 1 of the Response__
> >
> > > Choice of the measure M needs more analysis. This part is the key to this paper since the paper is built on a single intermediate density. This paper will be improved a lot if the authors can provide a detailed analysis and argument on how to choose a good measure M and even prove some useful theorems (a better convergence rate, an elegant representation, an efficient algorithm). However, the authors only mentioned three ideas without much analysis on each of them.
> >
> > There is another misunderstanding of sDRE here. Please note, we use more than a ‘single’ M measure in almost all experiments. We introduced some new M measures on top of TRE and provided the condition ($p << m$ and $q << m$) under which new measures can be constructed. We rigorously evaluated all of these measures (Fig 2 Table 2, 3, 5). Our primary contribution is not these intermediate distributions, but the multi-class classification based sDRE which we have demonstrated to work with both our and TRE’s M measures. Exploring M measures and further analysis is beyond the scope of this work.
> >
> > > Estimation/Learning of η(x). It was never explicitly explained how the authors estimate the class-probability η(x). From the contents, I guess the authors apply a logistic regression method. But again, η(x) can be estimated by other approach such as a tree classifier. Is the logistic regression offers any benefits on the later analysis? Or can we use other generative classifiers? Questions like this should be answered.
> >
> > We agree that it might be possible to use other classifiers to model $\eta$, however, we propose sDRE method as a multi-class logistic regression for $\eta$. Logistic regression is one of the most successful classifier based DRE methods, and therefore we primarily used it in this work. This is a common practice in most of the recent works in this line (Gutmann & Hyv ̈arinen, 2010; Gutmann& Hirayama, 2011; Sugiyama et al., 2012; Rhodes et al, 2020, Choi et al, 2021). Exploring how other classifier based DREs may benefit from sDRE is interesting but beyond the scope of this work.
> >
> > > A bit of redundant of FOD and HOD. It was not clear to me why the authors introduce the concept of FOD and HOD. First, these concepts were not well-defined beyond Gaussian. Second, these concepts were not related to the proposed method and were only used in the experiments. Why not just call them 'mean-shift' Gaussian problems and `variance-shift' Gaussian problems?
> >
> > We disagree with the reviewer that the concepts of FOD and HOD only apply to Gaussians. FOD pertains to discrepancy relating to the first order statistics and HOD for any other higher order statistics for all kinds of distributions (see Table 5, Fig 4). We further distinguish these two types of discrepancies under the umbrella of density chasm problem because recent works in this field are only capable of solving one or the other. We show that our proposed method solves both types of problems much more effectively than the current state-of-the-art.
> >
> > ---
> >
> > __We hope that we have remedied the misunderstandings that the reviewer had about our primary contributions that initially led them to doubt the novelty and impact of our work. In light of this, hopefully, they will reconsider our work.__

---

### Official Review · Reviewer_rhLq · 2021-11-02

**Correctness:** 3
**Technical Novelty And Significance:** 3
**Empirical Novelty And Significance:** 3
**Recommendation:** 5
**Confidence:** 4

**Main Review:**

Main review:
- This paper attempts to contribute to solving an important problem in the context of DRE.
- The proposed method seems to be an extension of Rhodes et al. (2020), but has a novelty.
- The authors propose a novel concepts: FOD and HOD, which are also interesting and meaningful in this context.

Questions:
1. What is the data-generating process?
2. What is the definition of $D_f$? $D_f(a||b) = B_f(a, b| M)$ for some $M$?
3. What is $\hat{\eta}$ in the denominator of $\hat{r}(x) = \frac{\hat{\eta}(x)}{\hat{\eta}}$?
4. Can the authors compare their proposed method with more methods, such as plain DRE methods and the method proposed by Kato and Teshima (2021)?

Minor points:
1. I could not understand the following sentence:
> However, closed-form estimation of density ratios is impossible in most problems of interest, as it requires knowledge of the functional form of the underlying densities.
We can approximate the density ratio via a linear-in-parameter model with some kernel function. Combining with the linear-in-parameter model with certain methods yields an estimator with a closed form. For example, the uLISF has a closed form solution in the sense that we can compute the estimator analytically. In the other words, I could not catch the definition of "closed form."


**Summary Of The Paper:**

This paper addresses one of the essential tasks in machine learning, the problem of density ratio estimation (DRE). In this paper, the authors consider applying the binary classification method to DRE. When the samples are well separated, this problem becomes difficult. To tackle this problem, this paper proposes a framework called scaled Bregman divergence. Under this framework, the density ratio can be estimated successfully. The authors also present some applications in experiments.

**Summary Of The Review:**

This paper attempts to contribute to solving an important problem in the context of DRE. While the attempt is novel and very interesting, there is insufficient comparison with existing methods and justification of the proposed method. Therefore, I vote for weak reject. However, depending on the rebuttal, there is a possibility to raise this rating.

---

> ### Author Response · Authors · 2021-11-12
> **Response**
>
> We thank the reviewer again for their feedback and questions. We now address each of their concerns one by one.
>
> ---
>
> > What is the data-generating process?
>
> We do not assume that the data generating process is known to the model. However, if the question pertains to how we generate the data then please see the following:
> * High Dim/ Robustness: Section 5.2, first paragraph.
> * Representation Learning: Section 5.4, first paragraph.
>
> We ask the reviewers to kindly clarify if we have misunderstood their question.
>
> >What is the definition of Df? Df(a||b)=Bf(a,b|M) for some M?
>
> $D_f$ is Bregman Divergence (BD). Both proposition 1 and Lemma 1 use the relationship between BD and sBD to relate binary and multiclass classifiers to DRE respectively.
>
> > What is  η^ in the denominator of r^(x)=η^(x)η^?
>
> We apologize, this is a typo, it should have been $\hat \eta_C(x)$. That is, the probability of x under the class C.
>
> > Can the authors compare their proposed method with more methods, such as plain DRE methods and the method proposed by Kato and Teshima (2021)?
>
> We have added comparisons to Kato and Teshima (2021) in Appendix E, and in the 40-dimensional setting, it significantly fails to estimate the mutual information (estimated at 1.60 vs ground truth at 119.96).
>
> Could the reviewer please specify what they mean by “plain DRE methods”? Please note, for each of our experiments one of the baselines is always the vanilla binary classification based DRE (BC-DRE). To the best of our knowledge in high dimensional settings that we consider here, classification-based DREs tend to work best, as such we compared our work to the state-of-the-art in classifier-based DREs.
>
> > Minor points: I could not understand the following sentence:
> However, closed-form estimation of density ratios is impossible in most problems of interest, as it requires knowledge of the functional form of the underlying densities. We can approximate the density ratio via a linear-in-parameter model with some kernel function. Combining with the linear-in-parameter model with certain methods yields an estimator with a closed form. For example, the uLISF has a closed form solution in the sense that we can compute the estimator analytically. In the other words, I could not catch the definition of "closed form."
>
> We agree with the reviewer that “closed-form” estimators are possible. Our statement intended to imply the fact that closed-form solutions of the form $\hat{r}(x) = \hat{p}(x)/\hat{q}(x)$, where $\hat p$ and $\hat q$ are models /analytical expressions of densities $p$ and $q$ respectively, are hard to train/derive for interesting problems. We have amended it.
>
> ---
>
> __We hope that we have answered reviewers' questions sufficiently. As per their request, we have added comparisons to methods in Kato and Teshima (2021) and amended the notation and typos. We hope that they will consider raising their score.__

---

### Official Review · Reviewer_Cfja · 2021-11-03

**Correctness:** 4
**Technical Novelty And Significance:** 2
**Empirical Novelty And Significance:** 3
**Recommendation:** 6
**Confidence:** 3

**Main Review:**

Strength:

+ The method is clearly motivated, and is theoretically sound.

+ The paper is written clearly.

+ The proposed method shows significant advantages over other methods, especially when both FOD and HOD density-chasm issues are present.

Limitations/questions:

- The novelty of the proposed method seems incremental. Both Lemma 1 and the idea of telescoping are not new.

- Assume that two densities have disjoint support, then wouldn’t the density ratio estimation be a trivial problem as the ratio is always 0 (or infinity) ? Also, even though two target densities overlap, if one is a mean-shifted version of the other (and they are far apart), intuitively logistic regression would give a better classification due to better separation. Would this contradict the challenge of FOD that you claim? I may have misunderstood some setting of the problem so I hope the authors can help clarify it.

- Most experiments are on normal/Gaussian distributions. Very limited results are given on the robustness to other families of well-known distributions. I wonder how well the proposed method performs given more complicated distributions, e.g. mixture of Gaussian (as a single density) or other distributions of multiple modes.


Other comments:
On page 5, in the definition of Xm, x_p should be x_p^i



**Summary Of The Paper:**

This work estimates the ratio of two densities using intermediate densities that have sufficient overlap with the target densities. The proposed method is based on the scaled-Bregman Divergence and it can be equivalently formulated as a multi-class logistic regression problem. By combining the idea of telescoping, the proposed approach mitigate the issue of FOD/HOD density-chasm and show superior performance on a variety of datasets.

**Summary Of The Review:**

In summary, I like the idea of the paper, but most examples and experiments are on Gaussian distributions which may limit the scenarios of real applications. I would encourage the authors to include more results on more “malicious” densities, or at least explain why such suggested experiments do not make sense.

---

> ### Author Response · Authors · 2021-11-12
> **Response**
>
> We thank the reviewer again for their feedback and questions. We now address each of their concerns one by one.
>
> ---
>
> > The novelty of the proposed method seems incremental. Both Lemma 1 and the idea of telescoping are not new.
>
> There seems to be a misunderstanding of our main contribution. Neither Telescoping nor Lemma 1 are our contributions. Those are indeed prior work. The novel aspects of our approach are the following:
>
> * __Telescoping vs Scaling:__ sDRE takes a different approach to the telescoping than the one presented in Rhodes et al. (2020). Instead of computing a chain of K intermediate density ratios to estimate $p/q = p/m_1 \times m_1/m_2 \dots \times m_K/q$, sDRE proposes to scale the densities $p$ and $q$ by one of K different M measures. This allows for computing the log ratio as $\log p/q = \log p/m_i - \log q/m_i$ using any of the K M-measures instead of chaining them. This is explained in the second last paragraph of section 4.1.
>
> * __Multi-class vs Binary Classification:__ sDRE uses a multi-class classifier (with softmax cross entropy loss) as a DRE. As such, the log ratio between two distributions is represented by the difference of their corresponding logits (see section 4.1). This, using Lemma 1, is shown to not suffer from the distribution shift issues unlike the chain of binary classifier based DRE and leads to a drastic improvement in density ratio estimation compared to the state-of-the-art.
>
>
> > Assume that two densities have disjoint support, then wouldn’t the density ratio estimation be a trivial problem as the ratio is always 0 (or infinity) ?
>
> Yes, but we do not consider these trivial cases. Please see the truncated Gaussian example that we provided in Appendix G (Table 5) and Appendix B for a non-trivial example of partially overlapping distributions with finite support.
>
> > Also, even though two target densities overlap, if one is a mean-shifted version of the other (and they are far apart), intuitively logistic regression would give a better classification due to better separation. Would this contradict the challenge of FOD that you claim? I may have misunderstood some setting of the problem so I hope the authors can help clarify it.
>
> Yes, this is a misunderstanding. As explained in Rhodes et al (2020) and in our section 2.2, when the densities are easily separable, the classifier performs very well but the density ratio estimation suffers because many different boundaries can achieve equally good classification performance. This is what we refer to as the density chasm problem.
>
> > Most experiments are on normal/Gaussian distributions. Very limited results are given on the robustness to other families of well-known distributions. I wonder how well the proposed method performs given more complicated distributions, e.g. mixture of Gaussian (as a single density) or other distributions of multiple modes.
>
> To the best of our knowledge, we evaluate our method on more distributions than any of the baseline methods.
> In fact, we evaluated the method on many more distributions than Gaussians: We also used heavy-tailed distributions, finite-support distributions (Table 2, 5), as well as SpatialMultiOmniglot data (Fig 4) in our applications to representation learning. To clarify this, we have changed the title from “Robustness Evaluation” to “Robustness and Generalization Beyond Gaussian Distributions”.
> We now added an additional experiment with a mixture of Gaussians in Appendix G Table 5 (row 3). sDRE estimates the KL divergence to be $98.27$, when the ground truth is $105.60$.
>
> ---
>
> __We hope we have addressed the reviewers concern regarding the novelty of our approach and clarified that we have indeed evaluated our method on a variety of Gaussian and non-Gaussian datasets. We have updated the title of the relevant sections for clarity and we have added new results for the mixture of Gaussians.__

---

### Official Review · Reviewer_u7f6 · 2021-11-03

**Correctness:** 3
**Technical Novelty And Significance:** 3
**Empirical Novelty And Significance:** 3
**Recommendation:** 6
**Confidence:** 4

**Main Review:**

The paper is easy to read and generally well-written. However, parts of the paper seem inaccurate or not very rigorous.

After offering Lemma 1, the authors conclude that "Lemma 1 shows that a multi-class logistic regression leads to a multi-class density ratio estimator that minimizes the sBD..." However, Lemma 1 just shows the connection between regression and DRE, and does not really talk about "logistic regression". At least in the standard ML literature logistic regression is a linear classifier with logistic loss. Therefore I see no reason that logistic regression work, unless the distributions are nicely separated (like isotropic Gaussians with different means).

Also, in Lemma 1, $\hat{\eta}$ has not been defined properly. After the statement of Lemma 1, we read "....the reference density C". This is quite confusing, since C seems to be just the number of classes (and not a density). Moreover, $r(x)$ in the lemma is a number so how is $D(r(x)||\hat{r}(x))$ defined?

It will help if the authors formally define the problem of density ratio estimation, and its measure of success. The discussion of the proposed method in section 4 are at times vague.  It will help to have a clear algorithm, pseudocode, etc. rather than just explaining things.

I could not follow Section 4.3, in particular the argument that says replacing log(m/q) with -log(q/m) is useful. I also found the arguments like "TRE is only defined when p<<m<<q..." somewhat sloppy. Why is this true? What goes wrong if m=10q in some parts of the domain? Can you make the argument somewhat more formal? At the end of the same section, the authors make another argument about the benefits of using multiclass logistic regression. However, the connection between consistency of ranking-based noise constrastive estimation and the current paper is not really established. Therefore I don't find the argument well supported.


**Summary Of The Paper:**

Given samples from two distirbutions, how can we estimate the KL divergence between them? More generally, how can we estimate the ratio of the two densities at given points (e.g, on the samples from one of the dists)? The authors propose a novel method for density ratio estimation (DRE). DRE is challenging when the two distributions have different supports or the density ratio is very high. The authors propose a solution that builds on the previous work, namely Rhodes et al..,  which instead of computing the ratio directly, it writes it as multiplication of a series of density ratios between some intermediate distributions (telescoping).

The authors improve over Rhodes et al. by accounting for the distribution shift in a systematic way and also slightly changing the way the intermediate distributions are constructed. The empirical results are promising and show the benefit of the approach compared to the baselines.


**Summary Of The Review:**

The paper proposes a novel DRE method which addresses some of the issues that the previous methods (e.g., TRE) had. The paper is easy to read but some of the arguments are weak and not sufficiently justified. More formal arguments/statements will help with the accuracy and readability of the paper. The paper is not theoretically strong, but offers a useful and practical approach which can be impactful.

---

> ### Author Response · Authors · 2021-11-12
> **Response**
>
> We thank the reviewer again for their feedback and questions. We now address each of their concerns one by one.
>
> ---
>
> > After offering Lemma 1, the authors conclude that "Lemma 1 shows that a multi-class logistic regression leads to a multi-class density ratio estimator that minimizes the sBD..." However, Lemma 1 just shows the connection between regression and DRE, and does not really talk about "logistic regression". At least in the standard ML literature logistic regression is a linear classifier with logistic loss. Therefore I see no reason that logistic regression work, unless the distributions are nicely separated (like isotropic Gaussians with different means).
>
> This may be a misunderstanding of Lemma 1. Lemma 1 does NOT show “the connection between regression and DRE”. It shows how a multi-class classifier, with possibly nonlinear classification functions and not just linear ones, is connected to DRE.
>
> > Also, in Lemma 1, η^ has not been defined properly. After the statement of Lemma 1, we read "....the reference density C". This is quite confusing, since C seems to be just the number of classes (and not a density).
>
> Sorry about this typo. The $\eta$ in the denominator should be $\eta_C(x)$, which is the class-probability of X under C. We have updated the manuscript with the correction.
>
> > Moreover, r(x) in the lemma is a number so how is D(r(x)||r^(x)) defined?
>
> We apologize for this typo. $x$ should be $X$, the random variable. However, since $D_f$ is a Bregman Divergence, it can be used to measure discrepancies between two functions. We have updated the notation in the manuscript.
>
> > It will help if the authors formally define the problem of density ratio estimation, and its measure of success. The discussion of the proposed method in section 4 are at times vague. It will help to have a clear algorithm, pseudocode, etc. rather than just explaining things.
>
> We have added a formal description of the task of DRE in Section 2.
> We have used accuracy of KL divergence estimation as the measure of success to evaluate DREs in all the experiments.
>
> > I could not follow Section 4.3, in particular the argument that says replacing log(m/q) with -log(q/m) is useful. I also found the arguments like "TRE is only defined when p<<m<<q..." somewhat sloppy. Why is this true? What goes wrong if m=10q in some parts of the domain?
> Can you make the argument somewhat more formal?
>
> To alleviate the chance of potential misunderstanding, we would like to clarify that the notation $q<<m$ is used to denote that $q$ is absolutely continuous to $m$. We require this to ensure that the ratio q/m (and the Radon Nikodym derivative of the corresponding measures) exists and is well defined. Your example, $m=10q$ in some parts of the domain (or vice versa, $q = 10m$) is therefore not a problem as it will not violate $p << m << q$ (TRE) or $p << q << m$ (sDRE). We have an example of what actually happens in the case this requirement is not met in the Appendix B. Please also see Appendix G for an example of how sDRE is applied to distributions with finite support.
>
> > At the end of the same section, the authors make another argument about the benefits of using multiclass logistic regression. However, the connection between consistency of ranking-based noise constrastive estimation and the current paper is not really established. Therefore I don't find the argument well supported.
>
> As explained in Ma and Collins (2018), the difference between the ranking based NCE and binary NCE is in their losses. Binary NCE uses binary cross entropy (binary classification) and ranking-based NCE uses a softmax cross entropy (multi-class classification). Since sDRE uses the same softmax cross entropy loss, we can use the results from Ma and Collins (2018). We have added further clarification in the paper.
>
> ---
>
> __We apologize for the lack of clarity and notational typos that caused confusion. We have rectified all the inconsistencies now and hope that we have sufficiently clarified your concerns about the theoretical soundness of our work.__

---

> > ### Comment · Reviewer_u7f6 · 2021-11-30
> > **updated my score**
> >
> > Thanks for making an effort to address the concerns. I updated my score accordingly.
> >
> > I still think some of the points that you raised for answering reviewers' questions can be integrated better in the write up of the paper.

---

### Author Response · Authors · 2021-11-12
**General Response and Summary of Changes**

We thank all the reviewers for their feedback and questions. We realize that the main concerns about the theoretical rigour of our work stem from the fact that both Proposition 1 and Lemma 1 had typos and undefined terms that led to confusion. We apologize for these inconsistencies and have revised the manuscript to address each of these issues and improve the overall presentation. The following is the complete list of changes that we made.

### New experiments

1. Comparison to Kato & Tashima (2021): As suggested by Reviewer 3, we ran comparisons against Kato & Tashima (2021) in the high-dimensional mutual  Even in the 40-dimensional experiment from row 2 of Table 2, the method fails to correctly estimate the mutual information significantly (estimated at $1.60$ vs. ground truth at $119.96$).
2. Mixture of Gaussians: As suggested by Reviewer 2, we evaluated the robustness of our model in the case when $p$ is a 160-dimensional Gaussian distribution centered at $-1$ with the block diagonal covariance matrix as described in Section 5.2 and $q$ is a mixture of Gaussians centered at $0.9$ and $1.1$ with identity covariance matrices and included the results in Appendix G Table 5. We accurately estimate the KL divergence to be $98.27$ (ground truth is $105.60$). Further tests of robustness on finite support distributions and heavy-tailed distributions are also in the same table.

### Corrections to the manuscript

1. Updated titles for sections/subsections 2.1, 4.1 and paragraph under section 5.2.
2. Updated Lemma 1 notation and added definition of the missing terms.
    * Added missing definitions for $\pi_C$, $p_C$.
    * Corrected and defined $\eta_C$
    * Added clarification text.
3. Updated notation of Proposition 1.
    * x → X
    * Defined $D_f$
4. Formally defined the task of density ratio estimation at the end of section 2.
5. Added definition of Bregman Divergence in section 2.
6. Added clarification of the connection between sDRE and results from Ma and Collins (2018) at the end of section 4.1.
7. Added clarification to the statement regarding the "closed-form" DRE in the introduction section.
8. Added clarification to the density chasm problem in section 2.2.

---

### Author Response · Authors · 2021-11-22
**End of the initial discussion period**

As we approach the end of the initial discussion period and the final day that we can update the manuscript, we want to thank the reviewers again and humbly request to please let us know if they have any additional concerns and experiments that they would like us to address and/or add to the manuscript.

---

### Decision · Program_Chairs · 2022-01-20

**Decision:**

Reject

**Comment:**

The paper introduces a technique to improve density ratio estimation. This is an important problem and very relevant to the ICLR conference. The main idea is to consider density ratios with respect to intermediate distributions to “scale” the densities and make the ratios easier to estimate by training a suitable discriminative model (classifier). Reviewers found the idea interesting but there was a consensus the paper is not ready for publication.